# PrObeD: Proactive Object Detection Wrapper

**Vishal Asnani**
Michigan State University
asnanivi@msu.edu

**Abhinav Kumar**
Michigan State University
kumarab6@msu.edu

**Suya You**
DEVCOM Army Research Laboratory
suya.you.civ@army.mil

**Xiaoming Liu**
Michigan State University
liuxm@cse.msu.edu

## Abstract

Previous research in $2D$ object detection focuses on various tasks, including detecting objects in generic and camouflaged images. These works are regarded as passive works for object detection as they take the input image as is. However, convergence to global minima is not guaranteed to be optimal in neural networks; therefore, we argue that the trained weights in the object detector are not optimal. To rectify this problem, we propose a wrapper based on proactive schemes, PrObeD, which enhances the performance of these object detectors by learning a signal. PrObeD consists of an encoder-decoder architecture, where the encoder network generates an image-dependent signal termed templates to encrypt the input images, and the decoder recovers this template from the encrypted images. We propose that learning the optimum template results in an object detector with an improved detection performance. The template acts as a mask to the input images to highlight semantics useful for the object detector. Finetuning the object detector with these encrypted images enhances the detection performance for both generic and camouflaged. Our experiments on MS-COCO, CAMO, COD10K, and NC4K datasets show improvement over different detectors after applying PrObeD. Our models/codes are available at `https://github.com/vishal3477/Proactive-Object-Detection`.

## 1 Introduction

Generic $2D$ object detection (GOD) has improved from earlier traditional detectors [15, 20, 64, 65] to the deep-learning-based object detectors [8, 10, 26, 32, 52, 58]. Advancements in deep-learning-based methods underwent many architectural change over recent years, including one-stage [5, 43, 46, 52–54], two-stage [23, 24, 58], CNN-based [5, 14, 16, 21–23, 52, 54], transformer-based [8, 74], and diffusion-based [10] methods. All these methods aim to predict the $2D$ bounding box of the objects in the images and their category class.

Another emerging area related to generic object detection is camouflaged object detection [17, 18, 27–29, 34, 40] (COD). COD aims to detect and segment objects blended with the background [17, 18] via object-level mask supervision. Applications of COD include medical [19, 45], surveillance [11] and autonomous driving [69]. Early COD detectors exploit hand-crafted features [50, 61] and optical flow [33], while current methods are deep-learning-based. These methods utilize attention [9, 63], joint learning [40], image gradient [34], and transformers [48, 70].

All these methods take input images as is for the detection task and hence are called passive methods. However, there is a line of research on proactive methods for a wide range of vision tasks such as disruption [59, 60], tagging [68], manipulation detection [1], and localization [2]. Proactive methods

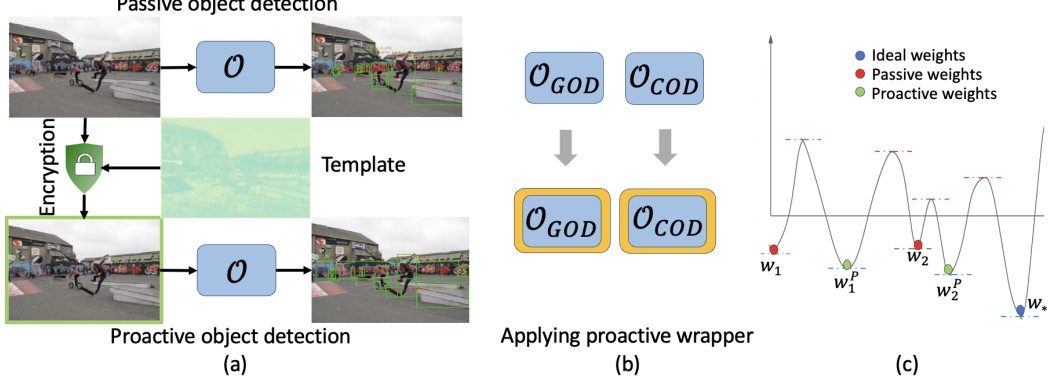

**Figure 1: (a) Passive *vs*. Proactive object detection**. A learnable template encrypts the input images, which are further used to train the object detector. **(b)** PrObeD serves as a wrapper on both generic and camouflaged object detectors, enhancing the detection performance. **(c)** For the linear regression model under additive noise and other assumptions, the converged weights of the proactive detector are closer to the optimal weights as compared to the converged weights of the passive detector. See Sec. 3.2 for details and proof.

use signals, called templates, to encrypt the input images and pass the encrypted images as the input to the network. These are trained in an end-to-end manner by using either a fixed [68] or learnable template [1, 2, 59, 60] to improve the performance. A major advantage of proactive schemes is that such methods generalize better on unseen data/models [1, 2]. Motivated by this, we propose a plug-and-play Proactive Object Detection wrapper, PrObeD, to improve GOD and COD detectors.

Designing PrObeD as a proactive scheme involves several challenges and key factors. First, the proactive wrapper needs to be a plug-and-play module that can be applied to both GOD and COD detectors. Secondly, the encryption process should be intuitive to benefit the object detection task. *e.g.*, an ideal template for detection should highlight the foreground objects in the input image. Lastly, the choice of supervision to estimate the template for encryption is hard to formulate.

Previous proactive methods [1, 2] use learnable but image-independent templates for manipulation and localization tasks. However, the object detection task is scene-specific; therefore, the ideal template should be image-dependent. Based on this key insight, we propose a novel plug-and-play proactive wrapper in which we apply object detectors to enhance detection performance. The PrObeD wrapper utilizes an encoder network to learn an image-dependent template. The learned template encrypts the input images by applying a transformation, defined as an element-wise multiplication between the template and the input image. The decoder network recovers the templates from the encrypted images. We utilize regression losses for supervision and leverage the ground-truth object map to guide the learning process, thereby imparting valuable object semantics to be integrated into the template. We then fine-tune the proactive wrapper with the GOD and COD detectors to improve their detection performance. Extensive experiments on MS-COCO, CAMO, COD10K, and NC4K datasets show that PrObeD improves the detection performance for both GOD and COD detectors.

In summary, the contributions of this work include:

- We propose a novel proactive approach *PrObeD* for the object detection task. To the best of our knowledge, this is the first work to develop a proactive approach to $2D$ object detection.
- We mathematically prove that the proactive method results in a better-converged model than the passive detector under assumptions and, consequently, a better object detector.
- PrObeD wraps around both GOD and COD detectors and improves detection performance on MS-COCO, CAMO, COD10K, and NC4K datasets

## 2 Related works

**Proactive Schemes.** Earlier works adopt to add signals like perturbation [60], adversarial noise [59], and one-hot encoding [68] messages while focusing on tasks like disruption [59, 60] and deepfake tagging [68]. Asnani *et al*. [1] propose to learn an optimized template for binary detection by unseen generative models. Recently, MaLP [2] adds the learnable template to perform generalized

**Table 1:** Comparison of PrObeD with prior works.

| Method | Proactive | Task | Template Number | Template Type | COD | GOD | Plug-Play |
|---|---|---|---|---|---|---|---|
| Faster R-CNN [58] | ✗ | Object Detection | - | - | ✗ | ✓ | ✗ |
| YOLO [52] | ✗ | Object Detection | - | - | ✗ | ✓ | ✗ |
| DeTR [8] | ✗ | Object Detection | - | - | ✗ | ✓ | ✗ |
| DGNet [34] | ✗ | Object Detection | - | - | ✓ | ✗ | ✗ |
| SINet-v2 [17] | ✗ | Object Detection | - | - | ✓ | ✗ | ✗ |
| JCSOD [40] | ✗ | Object Detection | - | - | ✓ | ✗ | ✗ |
| OGAN [60] | ✓ | Disrupt | 1 | Learnable | - | - | ✗ |
| Ruiz *et al*. [59] | ✓ | Disrupt | 1 | Learnable | - | - | ✗ |
| Yeh *et al*. [71] | ✓ | Disrupt | 1 | Learnable | - | - | ✗ |
| FakeTagger [68] | ✓ | Tagging | $\geq 1$ | Fixed, Id-dependent | - | - | ✗ |
| Asnani *et al*. [1] | ✓ | Manipulation Detection | $\geq 1$ | Learnable set, Image-independent | - | - | ✓ |
| MaLP [2] | ✓ | Manipulation Localization | $\geq 1$ | Learnable set, Image-independent | - | - | ✓ |
| PrObeD (Ours) | ✓ | Object Detection | $\geq 1$ | Learnable, Image-dependent | ✓ | ✓ | ✓ |

manipulation localization for unknown generative models. Unlike these works, PrObeD uses image-dependent templates and is a plug-and-play wrapper for a different task of object detection.

**Generic Object Detection** Detection of generic objects, instead of specific object categories such as pedestrians [7], apples [13], and others [4, 37, 38], has been a long-standing objective of computer vision. RCNN [24, 25] employs the extraction of object proposals. He *et al*. [31] propose a spatial pooling layer to extract a fixed-length representation of all the objects. Modifications of RCNN [23, 41, 58, 72] increase the inference speed. Feature pyramid network [42] detects objects with a wide variety of scales. The above methods are mostly two-stage, so inference is an issue. Single-stage detectors like YOLO [5, 52–54, 66], SSD [46], HRNet [67] and RetinaNet [43] increase the speed and simplicity of the framework compared to the two-stage detector. Recently, transformer-based methods [8, 74] use a global-scale receptive field. Chen *et al*. [10] use diffusion models to denoise noisy boxes at every forward step. PrObeD functions as a wrapper around the pre-existing object detector, facilitating its transformation into an enhanced object detector. The comparison of PrObeD with prior works is summarized in Tab. 1.

**Camouflaged Object Detection** Early COD works rely on hand-crafted features like co-occurrence matrices [61], $3D$ convexity [50], optical flow [33], covariance matrix [35], and multivariate calibration components [57]. Later on, [9, 63] incorporate an attention-based cross-level fusion of multi-scale features to recover contextual information. Mei *et al*. [49] take motivation by predators to identify camouflaged objects using a position and focus ideology. SINet [18] uses a search and identification module to perform localization. SINET-v2 [17] uses group-reversal attention to extract the camouflaged maps. [36] explores uncertainty maps and [75] utilizes cube-like architecture to integrate multi-layer features. ANet [39], LSR [47], and JCSOD [40] employ joint learning with different tasks to improve COD. Lately, [12, 48, 70] apply a transformer-based architecture for difficult-aware learning, uncertainty modeling, and temporal consistency. Zhai *et al*. [73] use a graph learning model to disentangle input into different features for localization. DGNet [34] uses image gradients to exploit intensity changes in the camouflaged object from the background. Unlike these methods, PrObeD uses proactive methods to improve camouflaged object detection.

# 3 Proposed Approach

Our method originates from understanding what makes proactive schemes effective. We first overview the two detection problems: GOD and COD in Sec. 3.1. We next derive Lemma 1, where we show that the proactive schemes with the multiplicative transformation of images are better than passive schemes by comparing the deviation of trained network weights from the optimal. Based on this result, we derive that Average Precision (AP) from the proactive model is better than AP from the passive model in Theorem 1. At last, we present our proactive scheme-based wrapper, PrObeD, in Sec. 3.3, which builds upon the Theorem 1 to improve generic 2D objects and camouflaged detection.

### 3.1 Background

#### 3.1.1 Passive Object Detection

Although generic $2D$ object detection and camouflage detection are similar problems, they have different objective functions. Therefore, we treat them as two different problems and define their objectives separately.

**Generic 2D Object Detection.** Let $\boldsymbol{I}_j$ be the set of input images given to the generic 2D object detector $\mathcal{O}$ with trainable parameters $\theta$. Most of these detectors output two sets of predictions per image: (1) bounding box coordinates, $\mathcal{O}(\boldsymbol{I}_j)_1 = \hat{T} \in \mathbb{R}^4$, (2) class logits, $\mathcal{O}(\boldsymbol{I}_j)_2 = \hat{C} \in \mathbb{R}^C$, where $N$ is the number of foreground object categories. If the ground-truth bounding box coordinates are $T_j$, and the ground-truth category label is $C$, the objective function of such detector is:

$$\min_{\theta} \left\{ \sum_j \left( ||\mathcal{O}(\boldsymbol{I}_j; \theta)_1 - T_j||_2 \right) - \sum_j \sum_{i=1}^{N} \left( C_j^i \cdot \log(\mathcal{O}(\boldsymbol{I}_j; \theta)_2) \right) \right\}. \tag{1}$$

**Camouflaged Object Detection.** Let $\boldsymbol{I}_j$ be the input image set given to the camouflaged object detector $\mathcal{O}$ with trainable parameters $\theta$, and $\boldsymbol{G}_j$ be the ground-truth segmentation map. Prior passive works predict a segmentation map with the following objective:

$$\min_{\theta} \left\{ \sum_j \left( \left\| \mathcal{O}(\boldsymbol{I}_j; \theta) - \boldsymbol{G}_j \right\|_2 \right) \right\}. \tag{2}$$

#### 3.1.2 Proactive Object Detection

Proactive schemes [1, 2] encrypt the input images with the template to aid manipulation detection/localization. Such schemes take an input image $\boldsymbol{I}_j \in \mathbb{R}^{H \times W \times 3}$ and learns a template $\boldsymbol{S}_j \in \mathbb{R}^{H \times W}$. PrObeD uses image-dependent templates to improve object detection. Given an input image $\boldsymbol{I}_j \in \mathbb{R}^{H \times W \times 3}$, PrObeD learns to output a template $\boldsymbol{S}_j \in \mathbb{R}^{H \times W}$, which can be used by a transformation $\mathcal{T}$ resulting in encrypted images $\mathcal{T}(\boldsymbol{I}_j)$. PrObeD uses element-wise multiplication as the transformation $\mathcal{T}$, which is defined as:

$$\mathcal{T}(\boldsymbol{I}_j) = \mathcal{T}(\boldsymbol{I}_j; \boldsymbol{S}_j) = \boldsymbol{I}_j \odot \boldsymbol{S}_j. \tag{3}$$

### 3.2 Mathematical Analysis of Passive and Proactive Detectors

PrObeD optimizes the template to improve the performance of the object detector. We argue that this template helps arrive at a better global minima representing the optimal parameters $\theta$. We now define the following lemma to support our argument:

**Lemma 1.** *Converged weights of proactive and passive detectors. Consider a linear regression model that regresses an input image $\boldsymbol{I}_j$ under an additive noise setup to obtain the 2D coordinates. Assume the noise under consideration $e$ is a normal random variable $\mathcal{N}(0, \sigma^2)$. Let $\boldsymbol{w}$ and $\boldsymbol{w}^*$ denote the trained weights of the pretrained linear regression model and the optimal weights of the linear regression model. Also, assume SGD optimizes the model parameters with decreasing step size $s$ such that the steps are square summable i.e., $\mathcal{S} = \lim_{t \to \infty} \sum_{k=1}^{t} s_k^2$ exist, and the noise is independent of the image. Then, there exists a template $\boldsymbol{S}_j \in [0, 1]$ for the image $\boldsymbol{I}_j$ such that the multiplicative transformation of images as the input results in a trained weight $\boldsymbol{w}'$ closer to the optimal weight than the originally trained weight $\boldsymbol{w}$. In other words,*

$$\mathbb{E}(||\boldsymbol{w}' - \boldsymbol{w}^*||_2) < \mathbb{E}(||\boldsymbol{w} - \boldsymbol{w}^*||_2). \tag{4}$$

The proof of Lemma 1 is in supplementary. We use the variance of the gradient of the encrypted images to arrive at this lemma. We next use Lemma 1 to derive the following theorem:

**Theorem 1.** *AP comparison of proactive and passive detectors. Consider a linear regression model that regresses an input image $\boldsymbol{I}_j$ under an additive noise setup to obtain the 2D coordinates. Assume the noise under consideration $e$ is a normal random variable $\mathcal{N}(0, \sigma^2)$. Let $\boldsymbol{w}$ and $\boldsymbol{w}^*$ denote the*

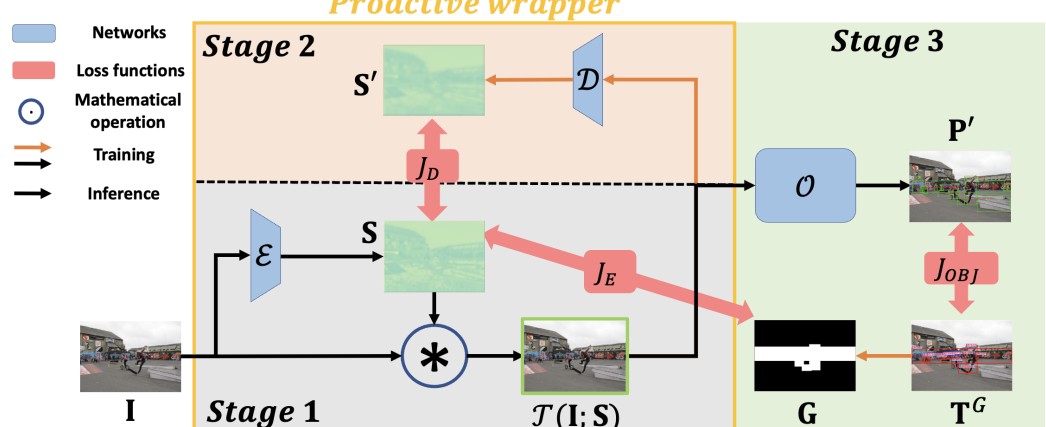

**Figure 2: Overview of PrObeD.** PrObeD consists of three stages: (1) template generation, (2) template recovery, and (3) detector fine-tuning. The templates are generated by encoder network $\mathcal{E}$ to encrypt the input images. The decoder network $\mathcal{D}$ is used to recover the template from the encrypted images. Finally, the encrypted images are used to fine-tune the object detector to perform detection. We train all the stages in an end-to-end manner. However, for inference, we only use stages 1 and 3. Best viewed in color.

*trained weights of the pretrained linear regression model and the optimal weights of the linear regression model. Also, assume SGD optimizes the model parameters with decreasing step size $s$ such that the steps are square summable i.e., $\mathcal{S} = \lim\limits_{t \to \infty} \sum\limits_{k=1}^{t} s_k^2$ exist, and the noise is independent of the image. Then, the AP of the proactive detector is better than the AP of the passive detector.*

The proof of Theorem 1 is in the supplementary. We use the Lemma 1 and the non-decreasing nature of AP w.r.t. IoU to arrive at this theorem. Next, we adapt the objectives of Eqs. (1) and (2) to incorporate the proactive methods as follows:

$$\min_{\theta, \boldsymbol{S}_j} \left\{ \sum_j \left( ||\mathcal{O}(\mathcal{T}(\boldsymbol{I}_j; \boldsymbol{S}_j); \theta)_1 - T_j||_2 \right) - \sum_j \sum_{i=1}^{N} \left( C_j^i \cdot \log(\mathcal{O}(\mathcal{T}(\boldsymbol{I}_j; \boldsymbol{S}_j); \theta)_2)) \right) \right\}, \quad (5)$$

$$\min_{\theta, \boldsymbol{S}_j} \left\{ \sum_j \left( \left\| \mathcal{O}(\mathcal{T}(\boldsymbol{I}_j; \boldsymbol{S}_j); \theta) - \boldsymbol{G}_j \right\|_2 \right) \right\}. \quad (6)$$

### 3.3 PrObeD

Our proposed approach comprises of three stages: template generation, template recovery, and detector fine-tuning. First, we use an encoder network to generate an image-dependent template for image encryption. This encrypted image is further used to recover the template through a decoder network. Finally, the object detector is fine-tuned using the encrypted images. All three stages are trained in an end-to-end fashion. While all the stages are used for training PrObeD, we specifically use only stages 1 and 3 for inference. We will now describe each stage in detail.

#### 3.3.1 Proactive Wrapper

Our proposed approach consists of three stages, as shown in Fig. 2. However, only the first two stages are part of our proposed proactive wrapper, which can be applied to object detector to improve its performance.

**Stage 1: Template Generation.** Prior works learn a set of templates [1,2] in their proactive schemes. This set of templates is enough to perform the respective downstream tasks as the generative model manipulates the template, which is easy to capture with a set of learnable templates. However, for object detection tasks, every image has unique object characteristics such as size, appearance, and color that can vary significantly. This variability present in the images may exceed the descriptive capacity of a finite set of templates, thereby necessitating the use of image-specific templates to

accurately represent the range of object features present in each image. In other words, a fixed set of templates may not be sufficiently flexible to capture the diversity of visual features across the given set of input images, thus demanding more adaptable, image-dependent templates.

Motivated by the above argument, we propose to generate the template $\boldsymbol{S}_j$ for every image using an encoder network. We hypothesize that highlighting the area of the key foreground objects would be beneficial for object detection. Therefore, for GOD, we use the ground-truth bounding boxes $T^G$ to generate the pseudo ground-truth segmentation map. Specifically, for any image $\boldsymbol{I}_j$, if the bounding box coordinates are $T_j^G = \{x_1, x_2, y_1, y_2\}$, we define the pseudo ground-truth segmentation map as:

$$\forall m \in [0, H], n \in [0, W], \text{ we have}$$
$$\boldsymbol{G}_j(m, n) = 1 \text{ if } x_1 \leq m \leq x_2 \text{ and } y_1 \leq n \leq y_2, \text{ otherwise } 0$$

However, for COD, the dataset already has the ground-truth segmentation map $\boldsymbol{G}_j$, which we use as the supervision for the encoder to output the templates with semantic information of the image to be restricted only in the region of interest for the detector. For both GOD and COD, we minimize the cosine similarity (Cos) between $\boldsymbol{S}_j$ and $\boldsymbol{G}_j$ as the supervision for the encoder network. The encoder loss $J_E$ is as follows:

$$J_E = 1 - \text{Cos}(\boldsymbol{S}_j, \boldsymbol{G}_j) = 1 - \text{Cos}(\mathcal{E}(\boldsymbol{I}_j), \boldsymbol{G}_j). \tag{7}$$

This generated template acts as a mask for the input image to highlight the object region of interest for the detector. We use this template with the transformation $\mathcal{T}$ to encrypt the input image as $\mathcal{T}(\boldsymbol{I}_j; \boldsymbol{S}_j) = \boldsymbol{I}_j \odot \boldsymbol{S}_j$. As we start from the pretrained model of object detector $\mathcal{O}$, we initialize the bias of the last layer of the encoder as 0 so that for the first few iterations, $\boldsymbol{S}_j \approx \boldsymbol{1}$. This is to ensure that the distribution of $\boldsymbol{I}_j$ and $\mathcal{T}(\boldsymbol{I}_j; \boldsymbol{S}_j)$ remains similar for the first few iterations, and $\mathcal{O}$ doesn't encounter a sudden change in its input distribution.

**Stage 2: Template Recovery.** So far, we have discussed the generation of template $\boldsymbol{S}_j$ using $\mathcal{E}$, which will be used as a mask to encrypt the input image. The encrypted images are used for two purposes: (1) recovery of templates and (2) fine-tuning of the object detector. The main intuition of recovering the templates is from the prior works on image steganalysis [55, 56] and proactive schemes [1, 2]. Motivated by these works, we draw the following insight: *"To properly learn the optimal template and embed it onto the input images, it is beneficial to recover the template from encrypted images."*

To perform recovery, we exploit an encoder-decoder approach. Using this approach leverages the strengths of the encoder network $\mathcal{E}$ for feature extraction, capturing the most useful salient details, and the decoder network $\mathcal{D}$ for information recovery, allowing for efficient and effective encryption and decryption of the template. We also empirically show that not using the decoder to recover the templates harms the object detection performance.

To supervise $\mathcal{D}$ in recovering $\boldsymbol{S}_j$ from $\mathcal{T}(\boldsymbol{I}_j; \boldsymbol{S}_j)$, we propose to maximize the cosine similarity between the recovered template, $\boldsymbol{S}_j'$ and $\boldsymbol{S}_j$. The decoder loss is as follows:

$$J_D = 1 - \text{Cos}(\boldsymbol{S}_j', \boldsymbol{S}_j) = 1 - \text{Cos}(\mathcal{D}(\mathcal{T}(\boldsymbol{I}_j; \boldsymbol{S}_j)), \boldsymbol{S}_j). \tag{8}$$

**Stage 3: Detector Fine-tuning.** Due to our encryption, the distribution of the images input to the pretrained $\mathcal{O}$ changes. Thus, we fine-tune $\mathcal{O}$ on the encrypted images $\mathcal{T}(\boldsymbol{I}_j; \boldsymbol{S})$. As proposed in Theorem 1, given the encrypted images $\mathcal{T}(\boldsymbol{I}_j; \boldsymbol{S})$, we use the pretrained detector $\mathcal{O}$ with parameters $\theta$ to arrive at a better local minima. Therefore, the general objective of GOD and COD in Eq. (5) and Eq. (6) change to as follows:

$$\min_{\theta, \theta_\mathcal{E}, \theta_\mathcal{D}} \left\{ \sum_j \left( ||\mathcal{O}(\mathcal{T}(\boldsymbol{I}_j; \mathcal{E}(\boldsymbol{I}_j; \theta_\mathcal{E})); \theta, \theta_\mathcal{D})_1 - T_j||_2 - \sum_{i=1}^N \left( C_j^i . \log(\mathcal{O}(\mathcal{T}(\boldsymbol{I}_j; \mathcal{E}(\boldsymbol{I}_j; \theta_\mathcal{E})); \theta, \theta_\mathcal{D})_2)) \right) \right) \right\}, \tag{9}$$

$$\min_{\theta, \theta_\mathcal{E}, \theta_\mathcal{D}} \left\{ \sum_j \left( \left\| \mathcal{O}(\mathcal{T}(\boldsymbol{I}_j; \mathcal{E}(\boldsymbol{I}_j; \theta_\mathcal{E})); \theta, \theta_\mathcal{D}) - \boldsymbol{G}_j \right\|_2 \right) \right\}. \tag{10}$$

We use the detector-specific loss function $J_{OBJ}$ of $\mathcal{O}$ along with the encoder and decoder loss in Eq. (7) and Eq. (8) to train all the three stages. The overall loss function $J$ to train PrObeD is as follows:

$$J = \lambda_{OBJ} J_{OBJ} + \lambda_E J_E + \lambda_D J_D. \tag{11}$$

**Table 2: GOD results** on MS-COCO val split. PrObeD improves the performance of all GOD at all thresholds and across all categories.

| Method | AP ↑ | AP$_{50}$ ↑ | AP$_{75}$ ↑ | AP$_S$ ↑ | AP$_M$ ↑ | AP$_L$ ↑ |
|---|---|---|---|---|---|---|
| Faster R-CNN [58] | 19.3 | 42.5 | 16.9 | 1.8 | 17.9 | 39.3 |
| Faster R-CNN [58]+PrObeD | **31.7** | **52.6** | **33.3** | **11.0** | **35.5** | **51.1** |
| Faster R-CNN + FPN [42] | 37.3 | 58.0 | 40.6 | 21.4 | 41.0 | 48.4 |
| Faster R-CNN + FPN [42] + Seg. Mask [30] | 38.2 | 60.3 | 41.7 | 22.1 | 43.2 | **51.2** |
| Faster R-CNN + FPN [42] + PrObeD | **38.5** | **60.4** | **41.9** | **22.5** | **43.4** | 49.8 |
| Sparse R-CNN [62] | 37.6 | 55.6 | 40.2 | 20.5 | 39.6 | 52.9 |
| Sparse R-CNN [62]+ PrObeD | **39.2** | **57.5** | **41.5** | **21.7** | **40.1** | **53.6** |
| YOLOv5 [52] | 48.9 | 67.6 | 53.1 | 31.8 | 54.4 | 62.3 |
| YOLOv5 [52]+ PrObeD | **49.4** | **67.9** | **53.5** | **32.0** | **55.1** | **62.6** |
| DeTR [8] | 41.9 | 62.3 | 44.1 | 20.3 | 45.8 | 61.0 |
| DeTR [8]+ PrObeD | **42.1** | **62.6** | **44.4** | **20.4** | **46.0** | **61.3** |

**Table 3: COD results** on CAMO, COD10K and NC4K datasets. PrObeD outperforms DGNet on all datasets and metrics.

| Method | CAMO | | | | COD10K | | | | NC4K | | | |
|---|---|---|---|---|---|---|---|---|---|---|---|---|
| | $E_m$↑ | $S_m$↑ | wF$_\beta$↑ | MAE↓ | $E_m$↑ | $S_m$↑ | wF$_\beta$↑ | MAE↓ | $E_m$↑ | $S_m$↑ | wF$_\beta$↑ | MAE↓ |
| DGNet [34] | 0.859 | 0.791 | 0.681 | 0.079 | 0.833 | 0.776 | 0.603 | 0.046 | 0.876 | 0.815 | 0.710 | 0.059 |
| + PrObeD | **0.871** | **0.797** | **0.702** | **0.071** | **0.869** | **0.803** | **0.661** | **0.037** | **0.900** | **0.838** | **0.755** | **0.049** |

## 4 Experiments

We apply PrObeD for two categories of object detectors: GOD and COD.

**GOD Baselines.** For GOD, we apply PrObeD on four detectors with varied architectures: two-stage, one-stage, and transformer-based detectors, namely, Faster R-CNN [58], YOLO [52], Sparse R-CNN, and DeTR [8]. We use these works as baselines for three reasons: (1) varied architecture types, (2) their increased prevalence in the community, and (3) varied timelines (from earlier to recent detectors). We use the PyTorch [51] code of the respective detectors for our GOD experiments and use the corresponding GODs as our baseline. For YOLOv5 and DeTR, we use the official repositories released by the authors; for Faster R-CNN, we use the public repository "Faster R-CNN.pytorch". For other GOD detectors, we use Detectron2 library as the pre-trained detector. We use the ResNet101 backbone for Faster R-CNN, Sparse R-CNN and DeTR, and CSPDarknet53 for YOLOv5.

**COD Baselines.** For COD, we apply PrObeD on the current SoTA camouflage detector DGNet [34] and use DGNet as our baseline. For all object detectors, we use the pretrained model released by the authors and fine-tune them with PrObeD. Please see the supplementary for more details.

**Datasets.** Our experiments use the MS-COCO 2017 [44] dataset for GOD, while we use CAMO [39], COD10K [17], and NC4K [47] datasets for COD. We use the following splits of these datasets:

- MS-COCO 2017 Val Split [44]: It includes 118,287 images for training and $5K$ for testing.
- COD10K Val Split [17]: It includes 4,046 camouflaged images for training and 2,026 for testing.
- CAMO Val Split [39]: It includes $1K$ camouflaged images for training and 250 for testing.
- NC4K Val [47]: It includes 4,121 NC4K images. We use it for generalization testing as in [34].

**Evaluation Metrics.** We use mean average precision average at multiple thresholds in $[0.5, 0.95]$ (AP) for GOD as in [44]. We also report results at threshold of 0.5 (AP$_{50}$), threshold of 0.75 (AP$_{75}$) and at different object sizes: small (AP$_S$), medium (AP$_M$), and large (AP$_L$). For COD, we use E-measure $E_m$, S-measure $S_m$, weighted F1 score $wF_\beta$ and mean absolute error $MAE$ as [34].

### 4.1 GOD Results

**Quantitative Results.** Tab. 2 shows the results of applying PrObeD on GOD networks. PrObeD improves the average precision of all three detectors. The performance gain is significant for Faster R-CNN. As Faster R-CNN is an older detector, it was at a worse minima to start with. PrObeD improves the convergence weight of Faster R-CNN by a significant margin, thereby improving the performance. We further experiment with two variations of Faster R-CNN, namely, Faster R-CNN +

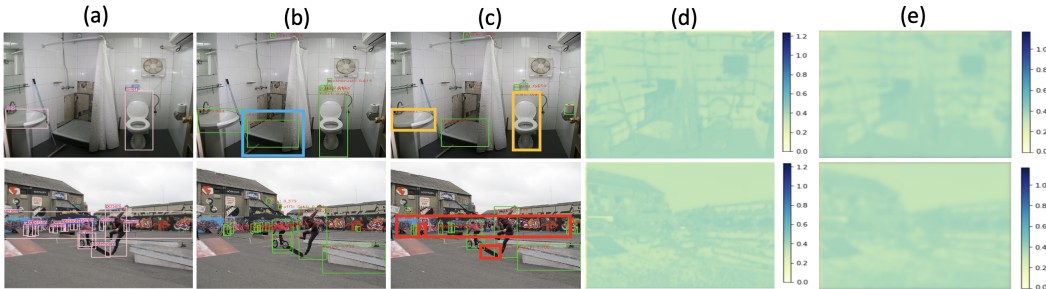

**Figure 3: Qualitative GOD Results** on MS-COCO 2017 dataset. (a) ground-truth annotations, (b) Faster R-CNN [58] predictions, (c) Faster R-CNN [58]+ PrObeD predictions, (d) generated template, and (e) recovered template. We highlight the objects responsible for improvement in (c) as compared to (b). The yellow box represents better localization, the blue box represents false positives, and the red box represents missed predictions. PrObeD improves on all these errors made by (b).

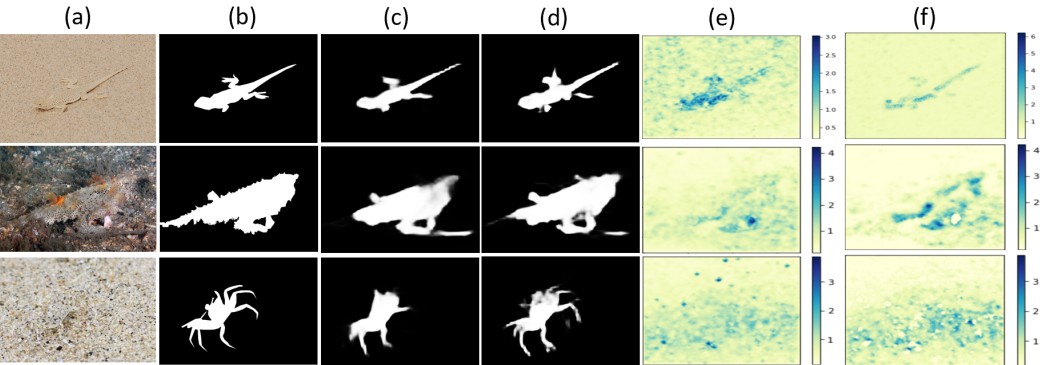

**Figure 4: Qualitative COD Results** on CAMO, COD10K, and NC4K datasets from top to bottom, after applying PrObeD. (a) input images, (b) ground-truth camouflaged map, (c) DGNet [34] predictions, (d) DGNet [34]+ PrObeD predictions, (e) generated PrObeD template, and (f) recovered PrObeD template. PrObeD template has the semantics of the camouflaged object, which aids DGNet in detection.

FPN and Sparse-RCNN. We observe an increase in the performance of both detectors. PrObeD also improves newer detectors like YOLOv5 and DeTR, although the gains are smaller compared to Faster R-CNN. We believe this happens because the newer detectors leave little room for improvement due to which PrObeD improves the performance slightly. We next compare PrObeD with a work that leverage segmentation map as a mask for object detection. We compare our performance with Mask R-CNN [30], which uses an image segmentation branch to help with object detection. Tab. 2 shows that the gains using Mask R-CNN are lower than using our proactive wrapper.

**Qualitative Results.** Fig. 3 shows qualitative results for the MS-COCO 2017 dataset. PrObeD clearly improves the performance of pretrained Faster R-CNN for three types of errors: Missed predictions, false negatives, and localization errors. PrObeD has a lower number of missed predictions, fewer false positives, and better bounding box localization. We also visualize the generated and recovered templates. We see that the template has object semantics of the input images. When the template is multiplied with the input image, it highlights the foreground objects, thereby making the task of object detector easier.

**Error Analysis.** We show the error analysis [6] for GOD section 4 of the supplementary. We observe that all GOD detectors make mistakes mainly due to five types of errors: classification, localization, duplicate detection, background detection, and missed detection. The main reason for the degraded performance is the errors in which the foreground-background boundary is missed. These errors include localization, background detection, and missed detection. Our proactive wrapper significantly corrects these errors, as the template has object semantics, which, when multiplied with the input image, highlights the foreground objects, consequently simplifying the task of object detection.

**Table 4:** Performance comparison with proactive works. MaLP [2] has a significantly deteriorated performance than PrObeD.

| Method | CAMO | | | | COD10K | | | | NC4K | | | |
|---|---|---|---|---|---|---|---|---|---|---|---|---|
| | $E_m\uparrow$ | $S_m\uparrow$ | $wF_\beta\uparrow$ | MAE$\downarrow$ | $E_m\uparrow$ | $S_m\uparrow$ | $wF_\beta\uparrow$ | MAE$\downarrow$ | $E_m\uparrow$ | $S_m\uparrow$ | $wF_\beta\uparrow$ | MAE$\downarrow$ |
| MaLP [2] | 0.474 | 0.514 | 0.218 | 0.254 | 0.491 | 0.520 | 0.150 | 0.202 | 0.503 | 0.548 | 0.228 | 0.222 |
| PrObeD | **0.871** | **0.797** | **0.702** | **0.071** | **0.869** | **0.803** | **0.661** | **0.037** | **0.900** | **0.838** | **0.755** | **0.049** |

**Table 5: Ablation studies** of PrObeD using Faster R-CNN GOD on MS-COCO 2017 dataset. Removing the encoder/decoder network or adding the template results in degraded performance.

| Changed | From→To | AP $\uparrow$ | $AP_{50}\uparrow$ | $AP_{75}\uparrow$ | $AP_S\uparrow$ | $AP_M\uparrow$ | $AP_L\uparrow$ |
|---|---|---|---|---|---|---|---|
| Template | Image Dependent→Fixed | 17.6 | 37.9 | 15.1 | 1.3 | 15.4 | 39.5 |
| | Image Dependent→Universal | 19.4 | 42.6 | 17.1 | 1.9 | 18.0 | 39.4 |
| Decoder | Yes→No | 25.2 | 46.1 | 26.2 | 5.3 | 26.6 | 24.1 |
| Transformation | Multiply→Add | 19.2 | 42.3 | 20.1 | 1.7 | 17.9 | 39.1 |
| PrObeD | - | **31.7** | **52.6** | **33.3** | **11.0** | **35.5** | **51.1** |

## 4.2 COD Results

**Quantitative Results.** Tab. 3 shows the result of applying PrObeD to DGNet [34] on three different datasets. PrObeD, when applied on top of DGNet, outperforms DGNet on all four metrics for all datasets. The biggest gain appears in COD10K and NC4K datasets. This is impressive as these datasets have more diverse testing images than CAMO. As NC4K is only a testing set, the higher performance of PrObeD demonstrates its superior generalizability as compared to DGNet [34]. This result agrees with the observation in [1, 2], where proactive-based approaches exhibit improved generalization on manipulation detection and localization tasks.

**Qualitative Results.** Fig. 4 visualizes the predicted camouflaged map for DGNet before and after applying PrObeD on testing samples of all three datasets. PrObeD improves the predicted camouflaged map, with less blurriness along the boundaries and better localization of the camouflaged object. As observed before for GOD, the generated and recovered template has the semantics of the camouflaged objects, which after multiplication intensifies the foreground object, resulting in better segmentation by DGNet.

## 4.3 Ablation Study

**Comparison with Proactive Works.** The prior proactive works perform a different task of image manipulation detection and localization. Therefore, these works are not directly comparable to our proposed proactive wrapper, which performs a different task of object detection as described in Tab. 1. However, manipulation localization and COD both involve a prediction of a localization map, segmentation, and fakeness map, respectively. This inspires us to experiment with MaLP [2] for the task of COD. We train the localization module of MaLP supervised with the COD datasets. The results are shown in Tab. 4. We see that MaLP is not able to perform well for all three datasets. MaLP is designed for estimating universal templates rather than templates tailored to specific images. It shows the significance of image-specific templates in object detection. While MaLP's design with image-independent templates is effective for localizing image manipulation, applying it to object detection has a negative impact on performance.

**Framework Design.** PrObeD consists of blocks to improve the object detector. Tab. 5 ablates different versions of PrObeD to highlight the importance of each block in our design. PrObeD utilizes an encoder network $\mathcal{E}$ to learn image-dependent templates aiding the detector. We remove the encoder $\mathcal{E}$ from our network, replacing it with a fixed template. We observe that the performance deteriorates by a large margin. Next, we make this template learnable as proposed in PrObeD, but only a single template would be used for all the input images. This choice also results in worse performance, highlighting that image-dependent templates are necessary for object detection. Finally, we remove the decoder network $\mathcal{D}$, which is used to recover the template from the encrypted images. Although this results in a better performance than the pretrained Faster R-CNN, we observe a drop as compared to PrObeD. Therefore, as discussed in Sec. 3.3, the recovery of templates is indeed a necessary and beneficial step for boosting the performance of the proactive schemes.

**Table 6: Ablation of training iterations** on Faster R-CNN. YOLOv5, and DeTR for more iterations similar to after applying PrObeD. We also report the inference time for all the detectors before and after applying PrObeD. Training object detectors proactively with PrObeD results in more performance gain compared to training passively for more iterations. PrObeD adds an overhead cost on top of the inference cost of detectors.

| Method | Iterations | AP ↑ | $AP_{50}$ ↑ | $AP_{75}$ ↑ | $AP_S$ ↑ | $AP_M$ ↑ | $AP_L$ ↑ | Time ($ms$) |
|---|---|---|---|---|---|---|---|---|
| Faster R-CNN [58] | 1× | 19.3 | 42.5 | 16.9 | 1.8 | 17.9 | 39.3 | 161.1 |
| Faster R-CNN [58] | 2× | 20.1 | 46.6 | 21.5 | 3.3 | 20.3 | 41.2 | |
| Faster R-CNN [58] + PrObeD | 2× | **31.7** | **52.6** | **33.3** | **11.0** | **35.5** | **51.1** | 175.3 (↑ 8.7%) |
| YOLOv5 [52] | 1× | 48.9 | 67.6 | 53.1 | 31.8 | 54.4 | 62.3 | 48.5 |
| YOLOv5 [52] | 2× | 48.8 | 67.7 | 53.0 | 31.8 | 54.7 | 62.4 | |
| YOLOv5 [52] + PrObeD | 2× | **49.4** | **67.9** | **53.5** | **32.0** | **55.1** | **62.6** | 62.7 (↑ 29.1%) |
| DeTR [8] | 1× | 41.9 | 62.3 | 44.1 | 20.3 | 45.8 | 61.0 | 194.2 |
| DeTR [8] | 2× | 41.9 | 62.4 | 44.0 | 20.1 | 45.9 | 61.1 | |
| DeTR [8] + PrObeD | 2× | **42.1** | **62.6** | **44.4** | **20.4** | **46.0** | **61.3** | 208.4 (↑ 7.2%) |

**Encryption Process.** PrObeD includes an encryption process as described in Eq. (3), which involves multiplying the template with the input image. This process makes the template act as a mask, highlighting the foreground for better detection. However, prior proactive works [1, 2] consider adding templates to achieve better results. Thus, we ablate by changing the encryption process to template addition. Tab. 5 shows that template addition degrades performance by a significant margin w.r.t. our multiplication scheme. This shows that encryption is a key step in formulating proactive schemes, and the same encryption process may not work for all tasks.

**More Training Time.** We perform an ablation to show that the performance gain of the detector is due to our proactive wrapper instead of training for more iterations of the pretrained object detector. Results in Tab. 6 show that although more training iterations for the detector has a performance gain, it's not enough to get the significant margin in performance as achieved by PrObeD. This shows that extra training can help, but only up to a certain extent.

**Inference Time.** We evaluate the overhead computational cost after applying PrObeD on different object detectors are shown in Tab. 6, averaged across $1,000$ images, on a NVIDIA $V100$ GPU. Our encoder network has 17 layers, which adds extra cost for inference. For detectors with bulky architectures like Faster R-CNN (ResNet101) and DeTR (transformer), the overhead computational cost is quite small, $8.7\%$ and $7.2\%$, respectively. This additional cost is minor compared to the performance gain of detectors, especially Faster R-CNN. For a lighter detector like YOLOv5, our overhead computational cost increases to $29.1\%$. So, there is a trade-off of applying PrObeD to different detectors with varied architectures. PrObeD is more beneficial to bulky detectors like two-staged/transformer-based as compared to one-stage detectors.

## 5 Conclusion

We mathematically prove that the proactive method results in a better-converged model than the passive detector under assumptions and, consequently, a better 2D object detector. Based on this finding, we propose a proactive scheme wrapper, PrObeD, which enhances the performance of camouflaged and generic object detectors. The wrapper outputs an image-dependent template using an encoder network, which encrypts the input images. These encrypted images are then used to fine-tune the object detector. Extensive experiments on MS-COCO, CAMO, COD10K, and NC4K datasets show that PrObeD improves the overall object detection performance for both GOD and COD detectors.

**Limitations.** Our proposed scheme has the following limitations. First, PrObeD does not provide a significant gain for recent object detectors such as YOLO and DeTR. Second, the proactive wrapper should be thoroughly tested on other object detectors to show the generalizability of PrObeD. Finally, we only experiment with simple multiplication and addition as the encryption scheme. A more sophisticated encryption process might further improve the object detectors' performance. We leave these for our future avenues.

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
