# PrObeD: Proactive Object Detection Wrapper
## – Supplementary material –

**Vishal Asnani**
Michigan State University
asnanivi@msu.edu

**Abhinav Kumar**
Michigan State University
kumarab6@msu.edu

**Suya You**
DEVCOM Army Research Laboratory
suya.you.civ@army.mil

**Xiaoming Liu**
Michigan State University
liuxm@cse.msu.edu

## 1   Proof of Lemma 1

We begin our proof by considering the image $i$ as a column vector and the model as a linear regression model with learnable weights $\boldsymbol{w}_t$. The subscript of time $t$ denotes that the weights change as one performs SGD updates.

**SGD Steps.**   We first consider the gradient of weight ($\boldsymbol{w}_t$). The linear model uses SGD for training, therefore, $\boldsymbol{w}_t$ after $t$ gradient steps is given by:

$$\boldsymbol{w}_t = \boldsymbol{w}_0 - \sum_{i=0}^{t} s_i \boldsymbol{g}_t = \boldsymbol{w}_0 - \sum_{i=0}^{t} s_i \frac{\partial \mathcal{L}}{\partial \boldsymbol{w}_t}, \tag{1}$$

where, for linear regression model with image $i$, $\mathcal{L} = f(\boldsymbol{w}_t i - z) = f(\eta)$. To estimate the gradient $\boldsymbol{w}_t$, we have,

$$\begin{aligned}
\boldsymbol{g}_t &= \frac{\partial \mathcal{L}(\boldsymbol{w}_t i - z)}{\partial \boldsymbol{w}_t} \\
&= \frac{\partial \mathcal{L}(\boldsymbol{w}_t i - z)}{\partial (\boldsymbol{w}_t i - z)} \frac{\partial (\boldsymbol{w}_t i - z)}{\partial \boldsymbol{w}_t} \\
&= \frac{\partial \mathcal{L}(\eta)}{\partial \eta} i \\
\boldsymbol{g}_t &= i\upsilon, \tag{2}
\end{aligned}$$

where $\upsilon = \frac{\partial \mathcal{L}(\eta)}{\partial \eta}$ is the gradient of the loss function wrt noise.

**Optimal Weights.**   First, we will find the bound of the converged value $\boldsymbol{w}_\infty$ and the optimal value $\boldsymbol{w}_*$. If $\mu_w$ is mean of the learned weight, we have,

$$\begin{aligned}
\mathbb{E}\left(\|\boldsymbol{w}_\infty - \boldsymbol{w}_*\|_2^2\right) &= \mathbb{E}\left(\|\boldsymbol{w}_\infty - \mu_w + \mu_w - \boldsymbol{w}_*\|_2^2\right), \\
&= \mathbb{E}((\boldsymbol{w}_\infty - \mu_w)^T(\boldsymbol{w}_\infty - \mu_w)) + \mathbb{E}((\mu_w - \boldsymbol{w}_*)^T(\mu_w - \boldsymbol{w}_*)) \\
&\quad + 2\mathbb{E}((\boldsymbol{w}_\infty - \mu_w)^T(\mu_w - \boldsymbol{w}_*)), \\
&= \mathbb{E}((\boldsymbol{w}_\infty - \mu_w)^T(\boldsymbol{w}_\infty - \mu_w)) + \mathbb{E}((\mu_w - \boldsymbol{w}_*)^T(\mu_w - \boldsymbol{w}_*)) \tag{3}
\end{aligned}$$

Using $\mathbb{E}(\boldsymbol{w}_\infty - \mu_w) = \mathbb{E}(\boldsymbol{w}_\infty) - \mu_w = \mu_w - \mu_w = 0$, we have

$$\implies \mathbb{E}\left(\|\boldsymbol{w}_\infty - \boldsymbol{w}_*\|_2^2\right) = Var(\boldsymbol{w}_\infty) + \mathbb{E}((\mu_w - \boldsymbol{w}_*)^T(\mu_w - \boldsymbol{w}_*)) \tag{4}$$

37th Conference on Neural Information Processing Systems (NeurIPS 2023).

where $Var(\boldsymbol{w}) = \sum_j w_j^2$.

**Gradient of Weight.** Given the image vector $\boldsymbol{i}$, and noise $\eta$ are statistically independent, the image and noise gradient $\upsilon$ defined in Eq. (2) are also statistically independent. We also assume that the distribution of image is normal Gaussian ($\mathbb{E}(\boldsymbol{i}) = 0$). Therefore, the expectation of the gradient $\boldsymbol{g}_t$ is given by,

$$\mathbb{E}(\boldsymbol{g}_t) = \mathbb{E}(\boldsymbol{i})\mathbb{E}(\upsilon) = 0, \tag{5}$$

Next, the variance of $\boldsymbol{g}_t$ is given as

$$Var(\boldsymbol{g}_t) = Var(\boldsymbol{i}\upsilon) = \mathbb{E}(\boldsymbol{i}^T\boldsymbol{i})[Var(\upsilon) + \mathbb{E}^2(\upsilon)] - \mathbb{E}(\boldsymbol{i})\mathbb{E}(\upsilon). \tag{6}$$

We assume that image pixels are normally distributed. This is common since the networks do a mean subtraction before inputting to the network. Thus, $\mathbb{E}(\boldsymbol{i}) = 0$. Hence, we have

$$Var(\boldsymbol{g}_t) = \mathbb{E}(\boldsymbol{i}^T\boldsymbol{i})Var(\upsilon). \tag{7}$$

**Converged Weight.** From Eq. (1), the expectation of the weight at time $t$ is,

$$\mathbb{E}(\boldsymbol{w}_t) = \mathbb{E}(\boldsymbol{w}_0) + \sum_{i=0}^{t} s_i \mathbb{E}(\boldsymbol{g}_j)$$
$$= 0 \text{ (Using Eq. (5))} \tag{8}$$

Therefore, for converged weight,

$$\mathbb{E}(\boldsymbol{w}_\infty) = \lim_{t\to\infty} \mathbb{E}(\boldsymbol{w}_t),$$
$$\mathbb{E}(\boldsymbol{w}_\infty) = \mathbb{E}(\mu_w) = 0. \tag{9}$$

For variance, using Eq. (1) we have,

$$Var(\boldsymbol{w}_t) = Var(\boldsymbol{w}_0) + (\sum_{i}^{t} s_j^2)Var(\boldsymbol{g}_t).$$

Therefore, we have,

$$Var(\boldsymbol{w}_\infty) = \lim_{t\to\infty}(Var(\boldsymbol{w}_t))$$
$$= Var(\boldsymbol{w}_0) + \left(\lim_{t\to\infty}\sum_{i=}^{t} s_j^2\right)Var(\boldsymbol{g}_t)$$
$$Var(\boldsymbol{w}_\infty) = Var(\boldsymbol{w}_0) + \mathcal{S}'Var(\boldsymbol{g}_t). \tag{10}$$

Substituting Eq. (7) in the above equation, we have

$$Var(\boldsymbol{w}_\infty) = Var(\boldsymbol{w}_0) + \mathcal{S}'\mathbb{E}(\boldsymbol{i}^T\boldsymbol{i})Var(\upsilon), \tag{11}$$

Going back to Eq. (4), and substituting Eq. (8) and Eq. (10), we have,

$$\mathbb{E}\left(\|\boldsymbol{w}_\infty - \boldsymbol{w}_*\|_2^2\right) = Var(\boldsymbol{w}_0) + \mathcal{S}'\mathbb{E}(\boldsymbol{i}^T\boldsymbol{i})Var(\upsilon) + \mathbb{E}(\|\boldsymbol{w}_*\|^2)$$
$$\implies \mathbb{E}\left(\|\boldsymbol{w}_\infty - \boldsymbol{w}_*\|_2^2\right) = c + \mathcal{S}Var(\upsilon) \tag{12}$$

where $c$ is independent of loss function $\mathcal{L}$ and $\mathcal{S} = \mathcal{S}'\mathbb{E}(\boldsymbol{i}^T\boldsymbol{i})$ is also another constant.

**Lemma** 1.

We assume that the regression error term $e = \boldsymbol{w}^T\boldsymbol{i} - \hat{y}$, is drawn from zero mean Gaussian with variance $\sigma^2$ as in [5]. So,

$$Var(\hat{e}) = Var(\boldsymbol{w}^T\boldsymbol{i} - \hat{y}) = \sigma^2. \tag{13}$$

For a passive detector with converged weights $\boldsymbol{w}_\infty$, we have,

$$\mathbb{E}\left(\|\boldsymbol{w}_\infty - \boldsymbol{w}_*\|_2^2\right) = c + \mathcal{S}Var(v)$$
$$= c + \mathcal{S}Var(e)$$
$$\implies \mathbb{E}\left(\|\boldsymbol{w}_\infty - \boldsymbol{w}_*\|_2^2\right) = c + \mathcal{S}\sigma^2 \tag{14}$$

Similarly, for a proactive detector with converged weights $\boldsymbol{w}'_\infty$, we have

$$\mathbb{E}\left(\left\|\boldsymbol{w}'_\infty - \boldsymbol{w}_*\right\|_2^2\right) = c + \mathcal{S}Var(v') \tag{15}$$

Assume that a proactive detector multiplies the input image vector $\boldsymbol{i}$ with a scalar template $s$. From Eq. (12), we write the loss term as,

$$\mathcal{L}' = \frac{1}{2}\left(s\boldsymbol{w}^T\boldsymbol{i} - \hat{y}\right)^2$$
$$\implies \frac{\partial \mathcal{L}'}{\partial \boldsymbol{w}} = (s\boldsymbol{w}^T\boldsymbol{i} - \hat{y})s\boldsymbol{i} \tag{16}$$

Taking the variance,

$$Var(v') = Var\left(\frac{\partial \mathcal{L}'}{\partial \boldsymbol{w}}\right) = Var((s\boldsymbol{w}^T\boldsymbol{i} - \hat{y})s\boldsymbol{i})$$
$$= Var(s(\hat{y} + e) - \hat{y})s^2 Var(\boldsymbol{i}) \quad \text{, assuming } \mathbb{E}(\boldsymbol{i}) = 0$$
$$= Var(se + (s-1)\hat{y})s^2 Var(\boldsymbol{i})$$
$$= (Var(se) + Var((s-1)\hat{y}))s^2 Var(\boldsymbol{i})$$
$$= s^2 Var(e)s^2 Var(\boldsymbol{i}) \quad \text{, assuming } Var(\hat{y}) = 0$$
$$\leq s^2 Var(e)s^2 \quad \text{, assuming } Var(\boldsymbol{i}) \leq 0.5 \times (-1)^2 + 0.5 \times 1^2 = 1 \tag{17}$$
$$\implies Var(v') \leq s^4 \sigma^2 \tag{18}$$

If the magnitude of the scalar template is bounded by 1 i.e., $s^2 < 1$, we have

$$Var(v') < \sigma^2. \tag{19}$$

The above shows that the gradients in the proactive model has less noise than the passive model (a key for better convergence). Substituting above in Eq. (15), we have

$$\mathbb{E}\left(\left\|\boldsymbol{w}'_\infty - \boldsymbol{w}_*\right\|_2^2\right) = c + \mathcal{S}Var(v')$$
$$< c + \mathcal{S}\sigma^2$$
$$< c + \mathcal{S}Var(v)$$
$$\implies \mathbb{E}\left(\left\|\boldsymbol{w}'_\infty - \boldsymbol{w}_*\right\|_2^2\right) < \mathbb{E}\left(\|\boldsymbol{w}_\infty - \boldsymbol{w}_*\|_2^2\right). \tag{20}$$

The last inequality follows trivially from Eq. (14).

## 2  Proof of Theorem 1

From Lemma 1, we have,

$$\mathbb{E}\left(\left\|\boldsymbol{w}'_\infty - \boldsymbol{w}_*\right\|_2^2\right) < \mathbb{E}\left(\|\boldsymbol{w}_\infty - \boldsymbol{w}_*\|_2^2\right)$$
$$\implies Var(\boldsymbol{w}'_\infty) < Var(\boldsymbol{w}_\infty)$$
$$\implies \mathbb{E}(|\boldsymbol{w}'^T_\infty\boldsymbol{i} - y|) < \mathbb{E}(|\boldsymbol{w}^T_\infty\boldsymbol{i} - y|)$$
$$\implies \mathbb{E}(\hat{y}' - y) < \mathbb{E}(\hat{y} - y) \tag{21}$$

Since the proactive detector has a better bounding box prediction,

$$\implies \mathbb{E}(IoU'_{2D}) > \mathbb{E}(IoU_{2D}) \tag{22}$$

Since $AP$ is a non-decreasing function of $IoU_{2D}$, we have,

$$AP^{\text{`}} \geq AP. \tag{23}$$

An important point to note is that the non-decreasing nature does not keep the inequality strict. In other words, we agree that the final AP from passive and pro-active schemes could be equal. However, our experience says that IoU improvements, especially close to $1$, lead to significant AP improvements. Current SoTA detectors already achieve decent IoU; hence, even a slight improvement in IoU improves the AP score.

**Table 1: Ablation of training iterations** on DGNet for more iterations similar to after applying PrObeD.

| Method | Iter | $E_m\uparrow$ | $S_m\uparrow$ | $wF_\beta\uparrow$ | MAE↓ | $E_m\uparrow$ | $S_m\uparrow$ | $wF_\beta\uparrow$ | MAE↓ | $E_m\uparrow$ | $S_m\uparrow$ | $wF_\beta\uparrow$ | MAE↓ |
|---|---|---|---|---|---|---|---|---|---|---|---|---|---|
| | | | CAMO | | | | COD10K | | | | NC4K | | |
| DGNet [6] | 1× | 0.859 | 0.791 | 0.681 | 0.079 | 0.833 | 0.776 | 0.603 | 0.046 | 0.876 | 0.815 | 0.710 | 0.059 |
| DGNet [6] | 2× | 0.861 | 0.791 | 0.682 | 0.080 | 0.832 | 0.778 | 0.606 | 0.045 | 0.875 | 0.814 | 0.711 | 0.059 |
| + PrObeD | 2× | **0.871** | **0.797** | **0.702** | **0.071** | **0.869** | **0.803** | **0.661** | **0.037** | **0.900** | **0.838** | **0.755** | **0.049** |

**Table 2:** Ablation of dice loss with cross-entropy (CE) loss *vs.* cosine similarity

| Method | CAMO | | | | COD10K | | | | NC4K | | | |
|---|---|---|---|---|---|---|---|---|---|---|---|---|
| | $E_m\uparrow$ | $S_m\uparrow$ | $wF_\beta\uparrow$ | MAE↓ | $E_m\uparrow$ | $S_m\uparrow$ | $wF_\beta\uparrow$ | MAE↓ | $E_m\uparrow$ | $S_m\uparrow$ | $wF_\beta\uparrow$ | MAE↓ |
| Dice + CE loss | 0.831 | 0.782 | 0.688 | 0.084 | 0.810 | 0.795 | 0.646 | 0.045 | 0.874 | 0.817 | 0.721 | 0.060 |
| Cosine similarity | **0.871** | **0.797** | **0.702** | **0.071** | **0.869** | **0.803** | **0.661** | **0.037** | **0.900** | **0.838** | **0.755** | **0.049** |

## 3 Implementation Details

We now include more details of our method here.

**Network Architecture.** The network architecture of encoder $\mathcal{E}$ and decoder $\mathcal{D}$ network used for PrObeD is shown in Fig. 1. Both networks consist of 2 stem convolution layers and 13 blocks, each block containing convolutional, batch normalization, and ReLU activation layers. The images are given as input to the encoder network to output the template, which is multiplied by the input images to make them encrypted. The encrypted images are then passed to the decoder network to recover the template. Finally, we input encrypted images to different object detectors to perform detection.

**Dataset license information.** We use benchmark datasets for GOD and COD. The authors for MS-COCO [8] dataset specify that the annotations in this dataset, along with this website, belong to the COCO Consortium and are licensed under a Creative Commons Attribution 4.0 License. The COD10K dataset is available for non-commercial purposes only [4]. The CAMO data is published under the Creative Commons Attribution-NonCommercial-ShareAlike 3.0 License [7]. Finally, the NC4K dataset is available to use for non-commercial purposes.

**Experimental Setup and Hyperparameters.** PrObeD is trained in an end-to-end manner for all the object detectors, with training iterations similar to the pretrained object detector. For both encoder and decoder networks, we use Adam optimizer with a learning rate of $1e^{-5}$. We use different weights of $[\lambda_{OBJ}, \lambda_E, \lambda_D]$ for different object detectors. We use [7,10,10] for Faster-RCNN, [50, 1.25, 4.25] for YOLOv5, [50, 7.5, 7.5] for DeTR and [10, 0.1, 0.1] for DGNet. All experiments are conducted on one NVIDIA A100 GPU.

## 4 Additional Experiments

**Train COD detector DGNet more.** Similar to the GOD detector, we train the COD detector DGNet for more iterations, similar to after applying PrObeD. The results are shown in Tab. 1. We see a similar behavior as seen in GOD detectors; the performance improves after training for more iterations, but only up to a certain extent. PrObeD is able to improve performance by a larger margin, showing the effectiveness of the proactive schemes.

**COD loss.** Our loss design is inspired by the prior proactive works [1, 2], which estimate the learnable template by applying a cosine similarity loss. The authors experiment with various loss types, showing the effectiveness of the cosine similarity loss design. However, COD is analogous to the segmentation task, which generally adopts a loss design of cross-entropy loss with dice loss, which might be beneficial for COD. We perform an ablation by applying cross-entropy loss with dice loss for COD. The results are shown in Table 2. We see that our proactive wrapper is not benefiting by removing the cosine similarity loss, proving the study of the prior proactive works.

**Error analysis.** Following [3], there can be a number of errors that deteriorate the performance of the object detector. These are:

1. Classification error (Cls): Localized correctly but classified incorrectly.

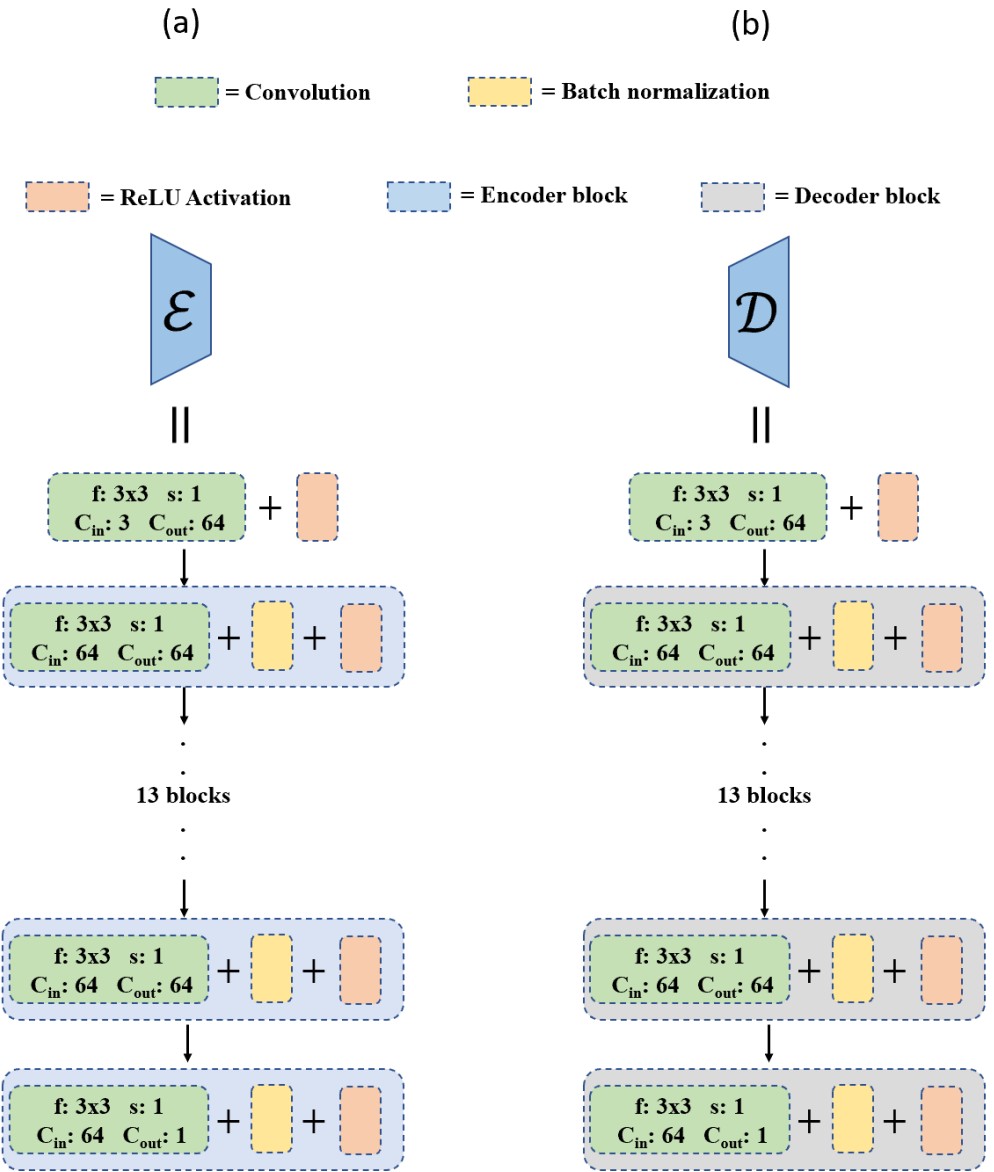

Figure 1: **Architecture** for encoder and decoder network.

2. Localization error (Loc): Classified correctly but localized incorrectly.

3. Both Classification and Localization error (Cls & Loc): Classified and localized incorrectly.

4. Duplicate detection error (Duplicate): Would be correct if not for a higher scoring detection.

5. Background error (Background): Detected background as foreground.

6. Missed target error (Missed): All undetected targets *i.e.*false negatives, which are not already covered by classification or localization errors.

Fig. 2 shows the error analysis for three object detectors, namely, Faster-RCNN, YOLOv5, and DeTR. PrObeD improves the number of correct predictions of all three detectors, especially for Faster-RCNN, where the number of correct predictions increases by around 17%. For DeTR and YOLOv5, the improvement is less, which is evident from the less increase in correct predictions. The major improvement for all three detectors comes from classification and localization-related errors. All these errors decrease after PrObeD is applied to all the detectors. Further, Faster-RCNN,

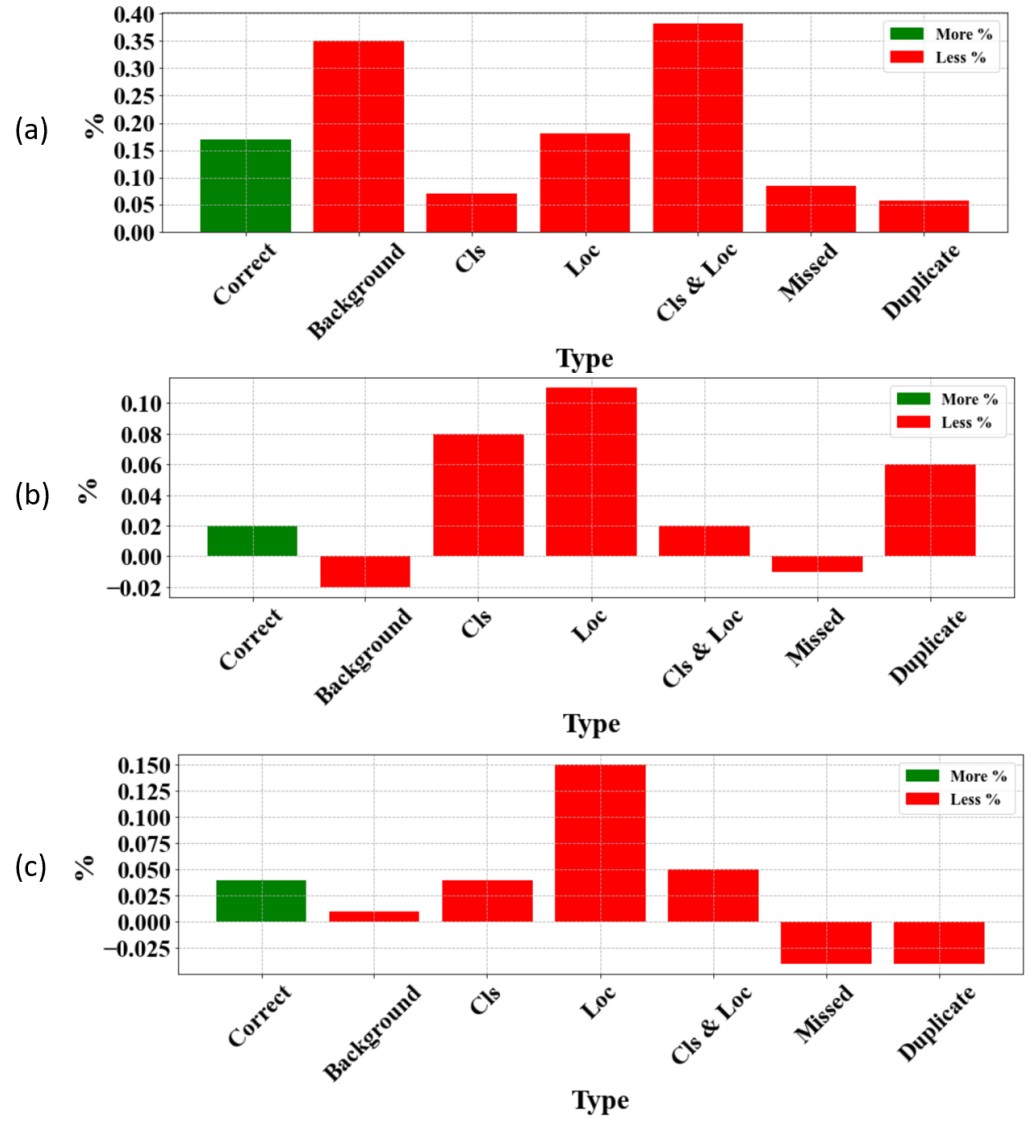

**Figure 2: Error analysis** for (a) Faster-RCNN, (b) YOLOv5, and (c) DeTR. PrObeD is able to improve the number of correct predictions and reduce most errors.

being an old detector, makes a lot of background errors, which are reduced by a significant margin after applying PrObeD. The gain is not much for DeTR and YOLOv5, which tend to make fewer background errors. Finally, one-stage detectors suffer mostly from the problem of duplicate detection, which is remedied by the PrObeD.

## 5 Potential Negative Societal Impact

PrObeD utilizes a proactive scheme to benefit object detection. Our approach can be considered a benign adversarial attack on object detectors. However, with a change in the objective function, PrObeD could also be used as an adversarial attack to deteriorate the performance of different object detectors. This might pose a threat to object detectors, whether used for GOD or COD, and some forms of adversarial training might be required to prevent the threat of adversarial attacks.