# OpenReview forum: "PrObeD: Proactive Object Detection Wrapper"
_NeurIPS.cc/2023/Conference — NeurIPS 2023 poster_

### Official Review · Reviewer_NMUy · 2023-06-28

**Soundness:** 2 fair
**Presentation:** 3 good
**Contribution:** 2 fair
**Rating:** 4
**Confidence:** 3

**Summary:**

The authors propose a proactive object wrapper for the object detection method. The warper is similar to an input transformation that makes the object detection model focus more on the salient regions. The experiments show that a consistent improvement after the proposed method is applied on top of existing objection methods.

**Strengths:**

The proposed method is somewhat novel in the sense that it is the first to use the proactive method on object detection tasks. Both GOD and COD situations are evaluated and the proposed method consistently improves the existing methods.

**Weaknesses:**

There are several weaknesses of the method, and some of them are also mentioned by the authors

- Performance gain: The performance gain becomes marginal when it comes to advanced object detection methods like YOLOv5 and DeTR. Considering two-stage methods, if Sparse R-CNN is evaluated instead of Faster R-CNN, would the improvement is also incremental?

- Simple transformation: The proactive scheme is a simple multiplication or addition between the input and the template. Is this process equivalent to having an image-dependent transformation that makes object detection focus on salient regions and improve performance? If yes, then I think there may be more different sophisticated techniques to be explored.

For the training time overhead, in table 5, it is mentioned the training time with and without PrObeD is reported. However, only the inference time is mentioned.

**Questions:**

See the Weakness.

**Limitations:**

See the Weakness.

---

> ### Author Rebuttal · Authors · 2023-08-09
>
> Thank you for your feedback. We provide a global response containing positives comments and major concerns of all reviewers. We also provide a global pdf file containing tables and figures with reference to added evaluation by some reviewers. We address all of your concerns below:
>
> $\textbf{Comparison with Sparse-RCNN}$
> That’s an excellent suggestion! We experiment with two variations of Faster-RCNN, namely, Faster-RCNN $+$ FPN and Sparse-RCNN (Faster-RCNN $+$ FPN $+$ Segmentation mask). The results are shown in Table $1$ of the global pdf file. We observe an increase in the performance for both the detectors, further proving the generalizability of our proactive wrapper.
>
> $\textbf{Image transformation.}$
> That is correct. Our proactive wrapper’s encryption scheme is an image-dependent transformation that makes object detectors focus on salient regions. We agree that there might be more sophisticated techniques to formulate the encryption scheme. We show that the idea of using a proactive wrapper on top of various object detectors (GOD/COD) increases the performance. As this is the first novel work to propose a proactive wrapper to improve object detection, we believe that PrObeD will motivate the future works to propose more sophisticated methods, which includes sophisticated encryption schemes to enhance the margin of improvement in the performance of object detection. Our error analysis in Figure 1 of supplementary proves the effectiveness of applying the idea of proactive schemes to improve the detection performance.
>
> $\textbf{Training/inference time overhead clarification. }$
> Thank you for pointing it out. We accept this mistake on our end. We meant to report the inference time overhead due to our proactive wrapper.

---

> > ### Comment · Reviewer_NMUy · 2023-08-11
> >
> > The rebuttal mainly answered my questions and I keep my score.

---

> > > ### Author Response · Authors · 2023-08-16
> > >
> > > Thank you for taking the time to go through our rebuttal.

---

### Official Review · Reviewer_rENj · 2023-06-29

**Soundness:** 3 good
**Presentation:** 3 good
**Contribution:** 2 fair
**Rating:** 4
**Confidence:** 3

**Summary:**

The paper presents a wrapper based on proactive object detection which can be utilized for both general object detection and camouflaged object detection problems. The proposed algorithm is flexible to different detection algorithms like FasterRCNN, Yolo and DeTR. Reasonable performance have been reported in benchmarks in general object detections like COCO and camouflaged object detection like CAMO, COD10K, and NC4K.

**Strengths:**

1. The proposed proactive object detection wrapper is flexible and can be applied to different object detectors.
2. The proposed algorithm has been verified on general object detection as well as camouflaged object detection benchmarks.

**Weaknesses:**

1. The motivation (as well as the advantages) of using the "template" is not clear.  As the generation of template relies on the ground-truth seg map, how about compairing the proposed algorithm with the algorithm which first perform segmentation to obtain the mask and then doing the object detection?

2. The performance gain for the general object detectors is not obvious, for example YOLOV5 and DeTR, both of which are not state-of-the-art object detectors. Especially, the proposed wrapper will introduce additional computational cost, which is not negligible.

3. The benchmark results on camouflaged object detection benchmarks (CAMO, COD10K, and NC4K) are slightly different from the results reported in other papers. For example, the baseline result reported in Table 3 is different from the results reported in DGNet[20]. Also, more comparison should be provided to validate the effectiveness of the proposed algorithm.

**Questions:**

Please address the questions raised in the weakness section, especially on the experimental results.

**Limitations:**

The authors have already provided the limitation discussion of the proposed algorithm.

---

> ### Author Rebuttal · Authors · 2023-08-09
>
> Thank you for your feedback. We provide a global response containing positives comments and major concerns of all reviewers. We also provide a global pdf file containing tables and figures with reference to added evaluation by some reviewers. We address all of your concerns below:
>
>
> $\textbf{Motivation of using the template.}$
> In our work, we propose to use a proactive scheme for improving the performance of object detection. As stated in the text, the major advantage of using the proactive schemes is for improving generalization. Prior works have proved this for different tasks such as image manipulation detection and localization. We aim to design a proactive scheme which is generalizable to different object detector, with an objective of increasing the performance. The learned template, which is the outcome of proactive learning, is multiplied with the input image. This can be seen as transforming the input data distribution to a perturbed data distribution which results in a better convergence and performance of the detector.
>
> $\textbf{Comparison with segmentation works.}$
> That’s a good point! It is beneficial to compare our method with the works that leverage segmentation map as a mask for object detection. We compare our performance with Mask R-CNN [He $\textit{et al.}$ in ICCV $2017$] which uses an image segmentation branch to help the object detection. Table 1 in global pdf file shows that the gains using Mask R-CNN are lower than using our proactive wrapper. Having said that, we also want to point out our baseline for COD, i.e. DGNET. The authors of DGNET leverage a gradient map as a mask to perform COD, and our proactive wrapper improves the performance of DGNET. This gradient map is analogous to using the segmentation maps as an extra information which is leveraged to perform the task of COD. Our wrapper is able to boost the performance of DGNet, highlighting the value of templates as extra information, and proactive schemes in general.
>
> $\textbf{Performance of YOLOv5 and DeTR.}$
> Thank you for your comment. We agree that the performance gain is not significant enough for YOLOv5 and DeTR as compared to Faster-RCNN. We argue that the new $2$D object detectors have very little room for improvement, therefore making it harder for PrObeD to improve the performance by a significant margin. We show that the idea of using a proactive wrapper on top of various object detectors (GOD/COD) will result in the increase in the performance. As this is the first novel work to propose a proactive wrapper to improve object detection, we believe that PrObeD will motivate the future works to propose more sophisticated wrappers to enhance the margin of improvement in the performance of object detection. Our error analysis in Figure $1$ of supplementary also proves the effectiveness of applying the idea of proactive schemes to improve the performance of the detectors.
>
> We will add this discussion in the manuscript.
>
>
> $\textbf{COD baseline performance.}$
> That’s a valid argument. We report the performance of the pretrained model released by the authors and use the same evaluation before and after applying PrObeD. Although the performance reported in the DGNET paper is higher than what is reported in our manuscript, we argue that it is very common to not have the exact replicated performance in object detection. Moreover, this issue has been raised by a lot of people in the publicly available repository, and the authors don’t comment on this issue. Further, we compare with the pretrained model released on the official repository and so, we believe our comparison is fair.

---

> > ### Comment · Reviewer_rENj · 2023-08-14
> >
> > The rebuttal addressed part of my questions. But I still have some concerns on the limited performance and the generalization of the proposed algorithm. Thus, I would maintain my score.

---

> > > ### Author Response · Authors · 2023-08-16
> > >
> > > Thank you for taking the time to go through our rebuttal.

---

### Official Review · Reviewer_93HY · 2023-07-02

**Soundness:** 2 fair
**Presentation:** 3 good
**Contribution:** 3 good
**Rating:** 6
**Confidence:** 4

**Summary:**

Authors propose a new object detection wrapper, PrObeD, based on proactive scheme, which tries to enhance the performance of existing object detectors. PrObeD generate a custom template for each input images which makes the weights of object detector closer to the ideal weights. Besides, a decoder who recovers the template from the encrypted images is presented to further boost the performance.

**Strengths:**

- PrObeD is the first work to develop a proactive approach upon object detection tasks, providing a new aspect to develop object detection models.
- Authors provide mathematical analysis of passive and proactive detectors, and prove that the proactive method results in a better-converged model than the passive detector under specific assumptions.
- Experiments show that PrObeD achieves SOTA performance on several GOD and COD datasets.

**Weaknesses:**

- The templates are applied on raw images with element-wise multiplication, which can be regarded as a soft mask. COD tasks may get benefit from it because it can mask useless background, but GOD tasks strongly dependent on image context, it's unclear why it's beneficial to GOD tasks.
- In Sec 3.2,  when the authors try to prove the rationality of template, they assume the object detector is a linear regression model, which is contrary to the factual circumstances.
- In line 191, authors said "it is beneficial to recover the template from encrypted images", but didn't provide explanations on it. Please provide the necessity or some intuitions behind this.

**Questions:**

see Weaknesses.

---

> ### Author Rebuttal · Authors · 2023-08-09
>
> Thank you for your feedback. We provide a global response containing positives comments and major concerns of all reviewers. We also provide a global pdf file containing tables and figures with reference to added evaluation by some reviewers. We address all of your concerns below:
>
>
> $\textbf{Benefit of adding Template for GOD task.}$
> That’s a good question! All the GOD detectors make mistakes mainly due to five types of errors as reported in the supplementary which are, classification, localization, duplicate detection, background detection, and missed detection. The main reason for performance to be reduced are the errors in which foreground-background boundary is missed. These errors include localization, background detection and missed detection. These errors are significantly corrected by our proactive wrapper, as the template has object semantics, which when multiplied with the input image highlights the foreground objects, thereby making the task of object detector easier. We hope this clears up the motivation of adding templates for GOD task.
>
> $\textbf{Proof assumptions}$
> A point to note is that mathematical proof and real world models do not share a one-to-one relationship. We must make assumptions to arrive at closed-form expressions to shed light on the problem we are trying to solve. While our analysis depends on these assumptions, it is significant that our results are evident even in real scenarios, even when the premises do not hold well. We leave the analysis after relaxing some or all of these assumptions for future avenues. Moreover, we believe our technical design and experimental results well justify our theoretical findings' effectiveness.
>
>
> $\textbf{Template recovery.}$
> Prior proactive works [Asnani $\textit{et al.}$ in CVPR 2022, Asnani $\textit{et al.}$ in CVPR 2023] on image manipulation detection and localization have showed that the template recovery is useful for learning the optimum template. Our design of proactive wrapper is inspired by these works. We back this claim by performing an ablation on removing the decoder responsible for template recovery from our wrapper. The ablation results are shown in main paper Table $4$. The performance decreases if the decoder is removed, proving the effectiveness of template recovery.

---

> > ### Comment · Reviewer_93HY · 2023-08-16
> > **Official Comment by Reviewer NMUy**
> >
> > Thanks for your reply. The rebuttal mainly answered my questions. However, I would like to see the authors explore more related work in the revised version, which would be more conducive to our understanding of the authors' contributions. For example, [ref1] "Camouflaged Object Detection with Feature Decomposition and Edge Reconstruction", [ref2] "Strategic Preys Make Acute Predators: Enhancing Camouflaged Object Detectors by Generating Camouflaged Objects", and [ref3] "Weakly-Supervised Concealed Object Segmentation with SAM-based Pseudo Labeling and Multi-scale Feature Grouping" could be applied to this challenging task.

---

> > > ### Author Response · Authors · 2023-08-16
> > >
> > > Thank you for your comment. We will surely discuss these works, and other related work (if any) in our revised version as advised. Thank you!

---

> > > > ### Comment · Reviewer_93HY · 2023-08-17
> > > > **Official Comment.**
> > > >
> > > > For me, I think the exploration done by the authors has some relevance to the field in question. Therefore I decided to raise the score to 6. Additionally, I strongly recommend the author discuss the differences between this work and the above-mentioned paper ([ref1-3]) to facilitate the reader's understanding.

---

> > > > > ### Author Response · Authors · 2023-08-17
> > > > >
> > > > > Thank you for increasing the rating. We agree with your suggestion. We have discussed the difference between our work and the above-mentioned papers ([ref1-3])  below:
> > > > >
> > > > > Ref1 performs the camouflaged object detection (COD) by decomposing the extracted features into different frequency bands using wavelet modules. Ref2 proposed a GAN-type framework where the generator and the detector are trained in an adversarial manner. Lastly, Ref3 leverages SAM for weakly-supervised segmentation to generate pseudo-labels to help detect concealed objects. All the above approaches are categorized as passive approaches as they take the input image as is. In contrast, our proposed proactive wrapper PrObeD performs camouflaged object detection (COD) and generic object detection (GOD) by leveraging an image-dependent template, which is used to transform the input images proactively to benefit the task of COD and GOD. PrObeD is not an object detector by itself; instead, it acts as a wrapper on different object detectors to boost the detection performance.
> > > > >
> > > > > We will add this discussion to the revised paper related works section.

---

> > > > > > ### Comment · Reviewer_93HY · 2023-08-17
> > > > > > **Official Comment.**
> > > > > >
> > > > > > I appreciate the efforts that the authors have made to address my main concerns. I think this paper is good. I would suggest submitting this paper to the arXiv regardless of the outcome, in order to move the field forward. Best of luck to the authors!

---

### Official Review · Reviewer_rzBs · 2023-07-04

**Soundness:** 3 good
**Presentation:** 1 poor
**Contribution:** 3 good
**Rating:** 5
**Confidence:** 3

**Summary:**

In this paper, the authors study the problem of proactive object detection. To be specific, the authors introduce a wrapper around an existing object detector that can produce a template image and improve the performance of the detector if the image is multiplied with this template. Prior works have proposed learning this template in an image independent manner. One of the important contributions of the paper is to make the template image dependent.

After the rebuttal:

I've read the other reviews and the responses provided by the authors. The authors have addressed my concerns and I have increased my recommendation. However, since the reported improvements are only minor, I feel unable to make a stronger recommendation.

**Strengths:**

1. Novel extension over prior work.
2. Significant improvements over baselines.

**Weaknesses:**

I don't work on proactive object detection. So, my evaluation maybe limited.

1. An important issue with the paper is that it fails to provide intuition on several design choices.  For example:

1.1. Lemma 1: The lemma is clear. But, what is the intuition for the template to improve the weights? Since the model is linear and the element-wise multiplication is linear, I don't see that adding S improves model capacity. The authors should provide some intuition here.

1.2. Why is template recovery helpful? Why is the recovered template too different from the generated one (in Fig 3)?

2. Theorem 1: "Then, the AP of the proactive detector is better than the AP of the passive detector" => They can be equal as well, if you check your proof carefully.

3. Experimental evaluation has major issues:

3.1. Certain crucial details are not provided:
  - It is not clear what framework (mmdetection, detectron, ..) is used.
  - Table 1: It is not clear what the backbones are.

3.2. Faster R-CNN's baseline performance (19.3 AP) is way too low compared to the known scores. See e.g. https://github.com/open-mmlab/mmdetection/tree/main/configs/faster_rcnn

3.3. No comparison is provided with the closest work [1,2].


Minor comments:

- "convergence to global minima is not optimal in neural networks" => Did you mean to say "not guaranteed" instead of "not optimal"?

- "weights in the object detector are not optimal" => "weights in object detectors are not optimal"?

- "schemes,PrObeD," => "schemes, PrObeD,".

- The abstract is overloaded with undefined unconventional terms (for the object detection literature) such as proactive scheme, template, ideal object detector... These make the abstract difficult to read.

- "has slowly improved from earlier traditional detectors" => I am not sure "slowly" is a suitable term here.

- Fig 1: The images are too small to comprehend what's being performed with the proposed method.

- Line 111: The symbol should be T for box coordinates.

- Eq 1: This is an overly simplistic formulation of the loss functions used in GOD. I would suggest using generic terms (such as L_box and L_cls) to keep the formulation general. Moreover, you should use \cdot instead of "." for multiplication (same for Eq 5).

- Eq 3: It is better to use \odot for element-wise multiplication.

- Proof of Lemma 1: It would be better to stick to the notation in the paper. E.g., for the template, you should use S instead of \alpha, and I for image instead of h.

- Fig 2: There are two arrows for training in the legend.

- Line 183: What is T here? The transformation or the box coordinates?

- "Faster-RCNN" => "Faster R-CNN".

**Questions:**

Please see above.

**Limitations:**

None.

---

> ### Author Rebuttal · Authors · 2023-08-09
>
> Thank you for your feedback. We provide a global response containing positives comments and major concerns of all reviewers. We also provide a global pdf file containing tables and figures with reference to added evaluation by reviewers. We address all of your concerns below:
>
> $\textbf{Clarification regarding Lemma 1.}$
> We want to clarify that our claim is not about model capacity. We only propose that using a template proactively leads to the optimum weights of object detector.
> Regarding the proof of lemma $1$, we will make the proof and text consistent in terms of variables for easy understanding.
>
> $\textbf{Clarification regarding Theorem 1.}$
> We agree that it could be equal since we state that AP is a non-decreasing function of IoU. However, our experience says that IoU improvements, especially close to $1$, lead to significant AP improvements.
> Current SoTA detectors already achieve decent IoU; hence, even a slight improvement in IoU improves the AP score.
>
> $\textbf{Motivation for template recovery.}$
> Prior proactive works [Asnani \etal in CVPR 2022, Asnani \etal in CVPR 2023] on image manipulation detection and localization show that the template recovery is useful for learning the optimum template. Our design of proactive wrapper is inspired by these works. We perform an ablation on removing the template recovery decoder from our wrapper. The ablation results in main paper Table $4$ show that the performance decreases if the decoder is removed.
>
> For Figure $3$, we apologize for overlooking this mistake. We came to know that the recovered template visualization in the manuscript is wrong. We remade the figure and show it in the global pdf file (Figure $1$).
>
> $\textbf{Experimental details.}$
> For YOLOv5 and DeTR, we use the official repositories released by authors. For Faster R-CNN, we use public repository "faster-rcnn.pytorch". We use the ResNet101 backbone for Faster R-CNN, CSPDarknet53 for YOLOV5, and ResNet101 for DeTR.
>
> $\textbf{Performance of Faster R-CNN.}$
> That’s an interesting point. We chose the original Faster R-CNN for our experiments. We report the results from the original Faster R-CNN paper [Ren $\textit{et al.}$ in NIPS $2015$] in our Table $2$ of paper. We agree that fine-tuning the model with optimal parameters and architecture selection boosts the performance. Having said that, we experiment with two variations of Faster R-CNN, namely, Faster R-CNN + FPN and Sparse-RCNN. The results are shown in Table $1$ of global pdf file. We observe an increase in the performance for both the detectors. This shows that PrObeD generalizes well to all the types of object detectors for increasing the performance. Also, the point to be noted here is that before and after applying our proposed PrObeD, the object detector is same in both cases. We show in the paper Table $5$ that even fine-tuning the detectors without our wrapper does increase the performance, but the gain is less compared to after applying PrObeD. Stating this, if any variation in the object detector hyperparameters or in architecture is helping the object detector, it will help even after applying PrObeD too.
>
> $\textbf{Comparison with proactive works.}$
> The prior proactive works perform a different task of image manipulation detection and localization. Therefore, these works are not directly comparable to our proposed proactive wrapper, which performs a completely different task of object detection as described in Table $1$ of the paper. However, manipulation localization and COD both involve a prediction of a localization map, segmentation and fakeness map respectively. This inspires us to experiment with MaLP (Asnani~\etal CVPR $23$) for the task of COD. We train the localization module of MaLP supervised with the COD datasets. The results are shown in Table $3$ in the global pdf file. We see that MaLP is not able to perform well for all three datasets. This was expected as MaLP is designed for image manipulation localization and extending it to object detection is not useful.
>
> $\textbf{Convergence to global minima.}$
> Great suggestion! Yes, we meant "not guaranteed" and we have fixed in paper.
>
> $\textbf{Abstract.}$
> We have updated the abstract to reduce and redefine some terms.
>
> Prior works in $2$D object detection focus on two different problems to detect objects in generic and camouflaged images. These works are regarded as passive works for object detection as they take the input image as is. However, convergence to global minima is not optimal in neural networks; therefore, we argue that the trained weights in the object detector are not optimal. To rectify this problem, we propose a wrapper based on proactive schemes, PrObeD, which enhances the performance of these object detectors by learning a signal. PrObeD consists of an encoder-decoder architecture, where the encoder network generates an image-dependent signals termed as templates to encrypt the input images, and the decoder recovers this template back from the encrypted images. We propose that learning the optimum template results in an object detector with an improved detection performance. The template acts as a mask to the input images to highlight semantics useful for the object detector. When the encrypted images are fine-tuned with the object detector, the detection performance for both generic and camouflaged object detectors enhances. Our experiments on MS-COCO, CAMO, COD$10$K, and NC$4$K datasets show improvement over different detectors on various datasets after applying PrObeD. Models and codes will be released upon publication.
>
> $\textbf{Figure 1.}$
> Thank you for pointing it out. We will incorporate your comment.
>
> $\textbf{Eq. $1$ and Eq. $3$.}$ Thank you for this suggestion! We will update these equations.
>
> $\textbf{Figure $2$.}$
> We want to specify that both orange and black arrow connected components are used for training. However, only black arrow components are needed for inference.
>
> $\textbf{Typos.}$
> We will update the text.

---

> > ### Comment · Reviewer_rzBs · 2023-08-13
> > **Re: Rebuttal by Authors**
> >
> > Thank you for the detailed rebuttal. Overall, I am happy with the responses provided.
> >
> > "We argue that the new 2D object detectors have very little room for improvement, therefore making it harder for PrObeD to improve the performance by a significant margin."
> >
> > I agree that this may be challenging. Though, papers to appear at venues like NeurIPS should be overcoming this challenge.
> >
> > I will increase my recommendation to reflect my views.

---

> > > ### Author Response · Authors · 2023-08-16
> > >
> > > Thank you for your comment. We are glad that you are satisfied with the rebuttal and have increased your recommendation. Thank you!

---

### Official Review · Reviewer_V1FT · 2023-07-09

**Soundness:** 3 good
**Presentation:** 3 good
**Contribution:** 3 good
**Rating:** 5
**Confidence:** 4

**Summary:**

The paper proposes a proactive approach to enhance the performance of object detection through adding a learnable image-dependent template into the image. The optimal template can effectively bridge the gap between the learned and the ideal weights for object detectors. As demonstrated by the results, the template acts as a mask of the input images to provide relevant information of the objects to guide the object detector for improved detection. The proposed approach can effectively improve the results of both generic and camouflaged object detection as shown in the extensive experimental results of diverse object detection benchmarks.

**Strengths:**

1. The proposed encoder-based learnable template can effectively highlight the semantic useful information to guide the object detector to perform improved detection.
2. The proposed learnable template can help significantly improve the performance of Faster-RCNN and other object detectors while not introducing much additional training and computational costs.
3. The authors have conducted extensive experimental evaluation to demonstrate their effectiveness.

**Weaknesses:**

1. For the generic object detection, the performance improvement for more modern object detectors, YOLOv5 and DeTR, is marginal and is not as significant as the Faster-RCNN. There is no discussions about this situation.
2. The performance of Faster-RCNN seems to be the old one. With proper parameter tuning and backbone selection as other modern object detectors, the performance should be much higher than 19.3, or even close to 40 on the MSCOCO dataset. The results of Faster-RCNN are not representative. However, most of the ablation studies of the proposed method on GOD are based on Faster-RCNN.
3. Some relevant works to leverage ground truth to generate a mask to help improve the object detection are not included for discussion in the related work.

Derakhshani, Mohammad Mahdi, Saeed Masoudnia, Amir Hossein Shaker, Omid Mersa, Mohammad Amin Sadeghi, Mohammad Rastegari, and Babak N. Araabi. "Assisted excitation of activations: A learning technique to improve object detectors." In Proceedings of the IEEE/CVF conference on computer vision and pattern recognition, pp. 9201-9210. 2019.

**Questions:**

After reading the proposed method, I think its functionality is close to a certain of region proposal generator to guide the object detector to better detect the objects. However, as shown in the Table 2, for one-stage detectors, YOLOv5 and DeTR both get marginal improvement. In contrast, the two-stage detector with the regional proposal network, Faster-RCNN gets significant performance improvement. It means that the proposed method can provide better object proposal information than the regional proposal network of Faster-RCNN. However, the performance of the original Faster-RCNN is very low as shown in the original NeurIPS paper. The updated Faster-RCNN with proper tuning and architecture selection can achieve much better performance. To consider these factors, I worry that the proposed method does not work as claimed for generic object detection. In addition, it is not clear for me why the authors choose to minimze the cosine similarity between the learnable template and the ground truth mask. I think the cross entropy loss with dice loss which are widely used in the segmentation task could be more appropriate to achieve the desired effect. Perhaps, the authors could further elaboration the loss design.

For camouflaged object detection, since it is formulated as a segmentation problem, it is reasonable that if the proposed method could provide object information, it will be very helpful for the following segmentation. However, I am curious about the comparisons of the region proposal network (if trained appropriately) and the proposed method in terms of the effectiveness and efficiency.

How do the authors actually conduct the element-wise multiplication between the template S_j and the input image I_j? The dimensions of both tensors seems mismatched where S_j is HxW but I_j is HxWx3.

**Limitations:**

Yes, the authors have addressed the limitation of their proposed method with sufficient context. For the negative societal impact, the authors could discuss more about avoiding the object detection from malicious monitoring use, etc.

---

> ### Author Rebuttal · Authors · 2023-08-09
>
> Thank you for your feedback. We provide a global response containing positives comments and major concerns of all reviewers. We also provide a global pdf file containing tables and figures with reference to added evaluation by some reviewers. We address all of your concerns below:
>
> $\textbf{Performance of YOLOv5 and DeTR.}$
> Thank you for your comment. We agree that the performance gain is not significant enough for YOLOv5 and DeTR as compared to Faster-RCNN. We argue that the new $2$D object detectors have very little room for improvement, therefore making it harder for PrObeD to improve the performance by a significant margin. We show that the idea of using a proactive wrapper on top of various object detectors (GOD/COD) will result in the increase in the performance. As this is the first novel work to propose a proactive wrapper to improve object detection, we believe that PrObeD will motivate the future works to propose more sophisticated wrappers to enhance the margin of improvement in the performance of object detection. Our error analysis in Figure $1$ of supplementary also proves the effectiveness of applying the idea of proactive schemes to improve the performance of the detectors.
>
> We will add this discussion in the manuscript.
>
>
> $\textbf{Performance of Faster-RCNN.}$
> That’s an interesting point. We chose the original Faster R-CNN for our experiments. We report the results from the original Faster R-CNN paper [Ren $\textit{et al.}$ in NIPS $2015$] in our Table $2$ of paper. We agree that fine-tuning the model with optimal parameters and architecture selection boosts the performance. Having said that, we experiment with two variations of Faster R-CNN, namely, Faster R-CNN + FPN and Sparse-RCNN. The results are shown in Table $1$ of global pdf file. We observe an increase in the performance for both the detectors. This shows that PrObeD generalizes well to all the types of object detectors for increasing the performance. Also, the point to be noted here is that before and after applying our proposed PrObeD, the object detector is same in both cases. We show in the paper Table $5$ that even fine-tuning the detectors without our wrapper does increase the performance across various object detectors including GOD and COD, but the gain is less compared to after applying PrObeD. Stating this, if any variation in the object detector hyperparameters or in architecture is helping the object detector, it will help even after applying PrObeD too.
>
> $\textbf{Related works to leverge ground truth.}$
> We will added the following related prior works which leverage ground truth to generate a mask for improving the object detection into our manuscript.
>
>
> $1.$ Mahdi $\textit{et al.}$ "Assisted excitation of activations: A learning technique to improve object detectors." In CVPR $2019$.
>
> $2.$ Fidler $\textit{et al.}$. Bottom-up segmentation for top-down detection. In CVPR $2019$.
>
> $3.$ Gidaris $\textit{et al.}$. Object detection via a multi-region and semantic segmentation-aware cnn model. In ICCV $2015$ .
>
> $4.$ He, Kaiming $\textit{et al.}$. "Mask r-cnn." In ICCV. $2017$.
>
> We will add more works related to the ones above in the manuscript.
>
> $\textbf{Ablation of using cross entropy loss with dice loss.}$
> Our loss design is inspired by the prior proactive works [Asnani $\textit{et al.}$ CVPR $2022$, Asnani $\textit{et al.}$ CVPR $2023$]  which estimate the learnable template by applying a cosine similarity loss. The authors experiment with various loss types, showing the effectiveness of cosine similarity loss design. However, we agree that a cross-entropy loss with dice loss inspired by segmentation works might be beneficial for object detection. Due to the time constraint, we perform an ablation of applying cross-entropy loss with dice loss for the task of object detection only for COD. The results are shown in Table $2$ of global pdf file. We see that our proactive wrapper is not benefiting by removing the cosine similarity loss, proving the study of the prior proactive works.
>
>
> $\textbf{Predicting object information.}$
> We agree that predicting the object information for COD can be useful. However, the dataset used by the COD works don’t have this label information to help train the detector for predicting this information. We propose PrObeD as a wrapper on top of the existing object detectors to improve the performance. We don’t change the design of the object detector in any form.
>
> Another point to be stated here is that PrObeD is able to reduce the classification and localization errors of some of the GOD object detectors as shown in the figure 1 of the supplementary. Improvement of the performance across both COD and GOD detectors show the generalizability of PrObeD, for both in terms of detection and classification of objects.
>
> $\textbf{Clarification regarding  element-wise multiplication of template and image}$
> Due to the template having only one channel as compared to the input image with three channels, we repeat the single channel information of the template thrice to match the dimensions of the input images.

---

> > ### Comment · Reviewer_V1FT · 2023-08-19
> >
> > Thanks for the rebuttal. I think the authors have addressed my concern, and I have raised my rating.

---

> > > ### Author Response · Authors · 2023-08-19
> > >
> > > Thank you for your comment and for increasing your rating.

---

### Author Rebuttal · Authors · 2023-08-09

$\textbf{Positives}$

We thank all the reviewers for their feedback regarding our work. All the reviewers have greatly appreciated our proposed work. The positives comments by the reviewers are mentioned below:



$1.$ Novelty of PrObeD (Reviewers rzBs, 93HY, NMUy).

$2.$ A new aspect to develop object detection models. (Reviewer 93HY)

$3.$ A strong mathematical analysis (Reviewer 93HY)

$4.$ Good performance on several GOD and COD detectors. (Reviewer V1FT, 93HY, rENj, NMUy)

$5.$ PrObeD is an effective approach to guide the object detector to perform improved detection. (Reviewers V1FT, rzBs)

$6.$ Extensive experimental evaluation to demonstrate the effectiveness of PrObeD. (Reviewers V1FT, 93HY, rENj)

$7.$ Important contribution of making the template image independent. (Reviewer rzBs)

$\textbf{Reviewer concerns}$

We carefully address each of the concerns of the reviewers concerns in their individual responses. However, we mention here some of main concerns of all the reviewers collectively providing a good summary of the major concerns of the reviewers.

$\textbf{Performance of Faster-RCNN. (Reviewer V1FT, rzBs, NMUy)}$
That’s an interesting point. We chose the original Faster R-CNN for our experiments. We report the results from the original Faster R-CNN paper [Ren $\textit{et al.}$ in NIPS $2015$] in our Table $2$ of paper. We agree that fine-tuning the model with optimal parameters and architecture selection boosts the performance. Having said that, we experiment with two variations of Faster R-CNN, namely, Faster R-CNN + FPN and Sparse-RCNN. The results are shown in Table $1$ of global pdf file. We observe an increase in the performance for both the detectors. This shows that PrObeD generalizes well to all the types of object detectors for increasing the performance. Also, the point to be noted here is that before and after applying our proposed PrObeD, the object detector is same in both cases. We show in the paper Table $5$ that even fine-tuning the detectors without our wrapper does increase the performance across various object detectors including GOD and COD, but the gain is less compared to after applying PrObeD. Stating this, if any variation in the object detector hyperparameters or in architecture is helping the object detector, it will help even after applying PrObeD too.


$\textbf{Performance of YOLOv5 and DeTR. (Reviewer V1FT, rENj)}$
Thank you for your comment. We agree that the performance gain is not significant enough for YOLOv5 and DeTR as compared to Faster-RCNN. We argue that the new $2$D object detectors have very little room for improvement, therefore making it harder for PrObeD to improve the performance by a significant margin. We show that the idea of using a proactive wrapper on top of various object detectors (GOD/COD) will result in the increase in the performance. As this is the first novel work to propose a proactive wrapper to improve object detection, we believe that PrObeD will motivate the future works to propose more sophisticated wrappers to enhance the margin of improvement in the performance of object detection. Our error analysis in Figure $1$ of supplementary also proves the effectiveness of applying the idea of proactive schemes to improve the performance of the detectors.

We will add this discussion in the manuscript.



$\textbf{Motivation for template recovery. (Reviewer rzBS, 93HY)}$
Prior proactive works [Asnani $\textit{et al.}$ in CVPR 2022, Asnani $\textit{et al.}$ in CVPR 2023] on image manipulation detection and localization have showed that the template recovery is useful for learning the optimum template. Our design of proactive wrapper is inspired by these works. We back this claim by performing an ablation on removing the decoder responsible for template recovery from our wrapper. The ablation results are shown in main paper Table $4$. The performance decreases if the decoder is removed, proving the effectiveness of template recovery.


$\textbf{Benefit of adding Template for GOD task. (Reviewer 93HY)}$
That’s a good question! All the GOD detectors make mistakes mainly due to five types of errors as reported in the supplementary which are, classification, localization, duplicate detection, background detection, and missed detection. The main reason for performance to be reduced are the errors in which foreground-background boundary is missed. These errors include localization, background detection and missed detection. These errors are significantly corrected by our proactive wrapper, as the template has object semantics, which when multiplied with the input image highlights the foreground objects, thereby making the task of object detector easier. We hope this clears up the motivation of adding templates for GOD task.



$\textbf{Comparison with segmentation works. (Reviewer rENj)}$
That’s a good point! It is beneficial to compare our method with the works that leverage segmentation map as a mask for object detection. We compare our performance with Mask R-CNN [He $\textit{et al.}$ in ICCV $2017$] which uses an image segmentation branch to help the object detection. Table 1 in global pdf file shows that the gains using Mask R-CNN are lower than using our proactive wrapper. Having said that, we also want to point out our baseline for COD, i.e. DGNET. The authors of DGNET leverage a gradient map as a mask to perform COD, and our proactive wrapper improves the performance of DGNET. This gradient map is analogous to using the segmentation maps as an extra information which is leveraged to perform the task of COD. Our wrapper is able to boost the performance of DGNet, highlighting the value of templates as extra information, and proactive schemes in general.

---

> ### Comment · Area_Chair_AKM7 · 2023-08-18
>
> Dear Authors: Thank you for your rebuttal. I have looked at it and the reviewer comments. I have contacted the reviewers who have not responded yet. I will take into consideration the rebuttal and the discussions in the final decision. Thank you.

---

> > ### Author Response · Authors · 2023-08-19
> >
> > Dear AC,
> > Thank you for your comment and for taking the time to go through the rebuttal and discussions with the reviewers. We look forward to your decision. Thank you.

---

### Decision · Program_Chairs · 2023-09-21

**Decision:**

Accept (poster)

**Comment:**

The paper introduces a "proactive" paradigm for object detection where the authors propose introducing a learnable template that guides the object detector towards semantically relevant parts of the image. They provide both theoretical and experimental analysis of their work. While the reviews were mixed, many of the comments were supportive of the work. Reviewers liked the novelty of the idea and the thorough analysis of the method. The negative reviewers were generally satisfied with the answers to their queries, but still questioned the performance improvement and generalizability of the approach. The AC looked at the paper and reviews and believes that the idea has enough novelty for the paper to be presented at NeurIPS. The evaluation is quite extensive and the AC did not feel that the authors should be penalized on that ground.